# Medullary tachykinin precursor 1 neurons promote rhythmic breathing

Jean-Philippe Rousseau[1], Andreea Furdui[1], Carolina da Silveira Scarpellini[1], Richard L Horner[2,3], Gaspard Montandon[1,3]*

[1]Keenan Research Centre for Biomedical Sciences. St. Michael's Hospital, Unity Health Toronto, Toronto, Canada; [2]Department of Physiology, Faculty of Medicine, University of Toronto, Toronto, Canada; [3]Department of Medicine, Faculty of Medicine, University of Toronto, Toronto, Canada

**Abstract** Rhythmic breathing is generated by neural circuits located in the brainstem. At its core is the preBötzinger Complex (preBötC), a region of the medulla, necessary for the generation of rhythmic breathing in mammals. The preBötC is comprised of various neuronal populations expressing neurokinin-1 receptors, the cognate G-protein-coupled receptor of the neuropeptide substance P (encoded by the tachykinin precursor 1 or *Tac1*). Neurokinin-1 receptors are highly expressed in the preBötC and destruction or deletion of neurokinin-1 receptor-expressing preBötC neurons severely impair rhythmic breathing. Although, the application of substance P to the preBötC stimulates breathing in rodents, substance P is also involved in nociception and locomotion in various brain regions, suggesting that *Tac1* neurons found in the preBötC may have diverse functional roles. Here, we characterized the role of *Tac1*-expressing preBötC neurons in the generation of rhythmic breathing in vivo, as well as motor behaviors. Using a cre-lox recombination approach, we injected adeno-associated virus containing the excitatory channelrhodopsin-2 ChETA in the preBötC region of *Tac1*-cre mice. Employing a combination of histological, optogenetics, respiratory, and behavioral assays, we showed that stimulation of glutamatergic or *Tac1* preBötC neurons promoted rhythmic breathing in both anesthetized and freely moving animals, but also triggered locomotion and overcame respiratory depression by opioid drugs. Overall, our study identified a population of excitatory preBötC with major roles in rhythmic breathing and behaviors.

**\*For correspondence:**
gaspard.montandon@utoronto.ca

**Competing interest:** The authors declare that no competing interests exist.

## Editor's evaluation

This important study expands our understanding of the neuronal composition and functions of the brainstem region called the preBötzinger complex (preBötC)- the site where the rhythm of breathing in mammals originates. The authors identify from gene expression mapping a small population of primarily excitatory neurons expressing the gene (Tac 1) for Substance P and also, in many cells, co-expressing the mu-opioid receptor gene (Oprm1) within this region. From targeted optogenetic photostimulation studies in anesthetized and freely-behaving mice in vivo, the authors present solid evidence that the Tac1-expressing neurons play an important role in modulating the inspiratory rhythm, are capable of overcoming respiratory depression by opioid drugs (fentanyl in these studies), and also can induce locomotion.

## Introduction

Motor rhythms are fundamental for many biological functions including locomotion and breathing. Breathing relies on a respiratory network located in the brainstem to promote gas exchange and maintain life. Although these neural circuits are critical for life, they can be adjusted to coordinate

with behaviors such as pain response and vocalization. At the core of the respiratory network is the preBötC, a collection of neurons essential to produce and sustain breathing (*Smith et al., 1991*). The preBötC region is comprised of diverse populations of neurons encompassing inhibitory and excitatory neurons. Glutamatergic excitatory neurons form roughly half of the preBötC neurons with critical roles in the generation of inspiration. Excitatory preBötC neurons express various proteins, receptors, and neuropeptides, including the transcription factor developing brain homeobox 1 protein (Dbx1), vesicular-glutamate transporter 2 (*Slc17a6*), neurokinin-1 receptors (encoded by the tachykinin 1 receptor gene *Tacr1*), μ-opioid receptors (μOR), and somatostatin (*Gray et al., 1999*; *Stornetta et al., 2003*; *Bouvier et al., 2010*; *Gray et al., 2010*). A subpopulation of glutamatergic neurons expresses the neuropeptide substance P (encoded by the tachykinin precursor 1 or *Tac1* gene). Substance P in the preBötC stimulates breathing through activation of its cognate receptor neurokinin-1 receptors (*Montandon et al., 2016a*), and the destruction of neurokinin-1 receptor-expressing neurons abolishes breathing (*Gray et al., 2001*). However, the biological function of preBötC *Tac1* (substance P)-expressing neurons in the regulation of breathing is not known in vivo.

Substance P plays an important role in rhythmic breathing, but it is also a key-molecule involved in nociception (*Mantyh, 2002*), locomotion (*Farrell et al., 2021*), and arousal (*Reitz et al., 2021*). Substance P is released in response to nociceptive stimuli (*Mantyh, 2002*), is expressed in circuits regulating nociception such as the dorsal horn of the spinal cord (*Chang et al., 2019*) and the rostral ventromedial medulla, and modulates descending pain circuits (*Khasabov et al., 2017*). Descending *Tac1* circuits in the brainstem mediate behavioral responses, such as the fight-or-flight response, associated with brisk locomotor activity (*Barik et al., 2018*; *Kuwaki, 2021*). To anticipate the body's metabolic demand in the event of a locomotor nocifensive response, nociceptive stimuli elicit cardiorespiratory responses, such as increased heart rate and augmented breathing (*Jafari et al., 2017*). *Tac1*-expressing medullary circuits involved in nociception or breathing share similar properties: they are sensitive to opioid drugs and expressed neurokinin-1 receptors. Here, we aim to identify the role of *Tac1*-expressing preBötC cells in regulating breathing and motor behaviors, which constitute different components potentially linked to nocifensive behaviors.

By combining optogenetics, respiratory, and locomotion assays, we determined the role of *Tac1* preBötC cells in regulating rhythmic breathing and locomotion. Using photostimulation, we first showed that *Tac1* preBötC neurons, a subpopulation of glutamatergic neurons, increased respiratory rhythm by promoting inspiration or reducing expiration in both anesthetized and freely behaving mice. In freely moving mice, photostimulation was mostly effective when mice were in calm, but not active, state. Interestingly, stimulation of *Tac1* preBötC cells directly elicited a strong locomotor response suggesting that *Tac1* preBötC cells play a dual role in promoting breathing and locomotion. Rhythmic breathing is dampened by opioid drugs and preBötC cells mediate a major component of respiratory depression by opioid drugs (*Montandon et al., 2011*; *Montandon, 2022b*). Because most *Tac1* preBötC cells co-expressed μ-opioid receptors (encoded by the gene *Oprm1*) in the preBötC, their stimulation reversed the effects of opioid drugs on breathing, suggesting that *Tac1* preBötC neurons constitute a robust excitatory neuronal population promoting breathing during calm states and that can overcome respiratory depression with narcotics.

## Results

### Stimulation of *Tac1*-expressing cells in anesthetized mice

Using a Cre-lox recombination-based approach, we expressed ChETA (*Figure 1a*) in the medulla of *Tac1* cre mice. A series of five photostimulations was produced containing frequencies ranging between 5, 10, 20, 30, and 40 Hz (*Figure 1b*). As no changes in respiratory rate were observed in animals which received the virus 2 weeks prior to the experimental day (*Figure 1c*), experiments were also performed on animals 4 weeks following virus injections. In this group, laser stimulation increased respiratory rate compared to the control group (*Figure 1b and c*; group effect: p=0.0003, n=25; $F_{(2,22)}=11.68$). Absolute values for respiratory rate are provided in *Figure 1—figure supplement 1*. Diaphragm respiratory amplitude was unaffected by laser stimulation (*Figure 1d*; group effect: p=0.3606, n=23; $F_{(2,20)}=1.074$). Inspiratory duration was not influenced by stimulation (*Figure 1e*; group effect: p=0.0934, n=14; $F_{(1,12)}=3.321$) but expiratory duration decreased compared to the control group (*Figure 1f*; group effect: p=0.1158, n=14; $F_{(1,12)}=2.874$). Laser stimulations at 30 Hz

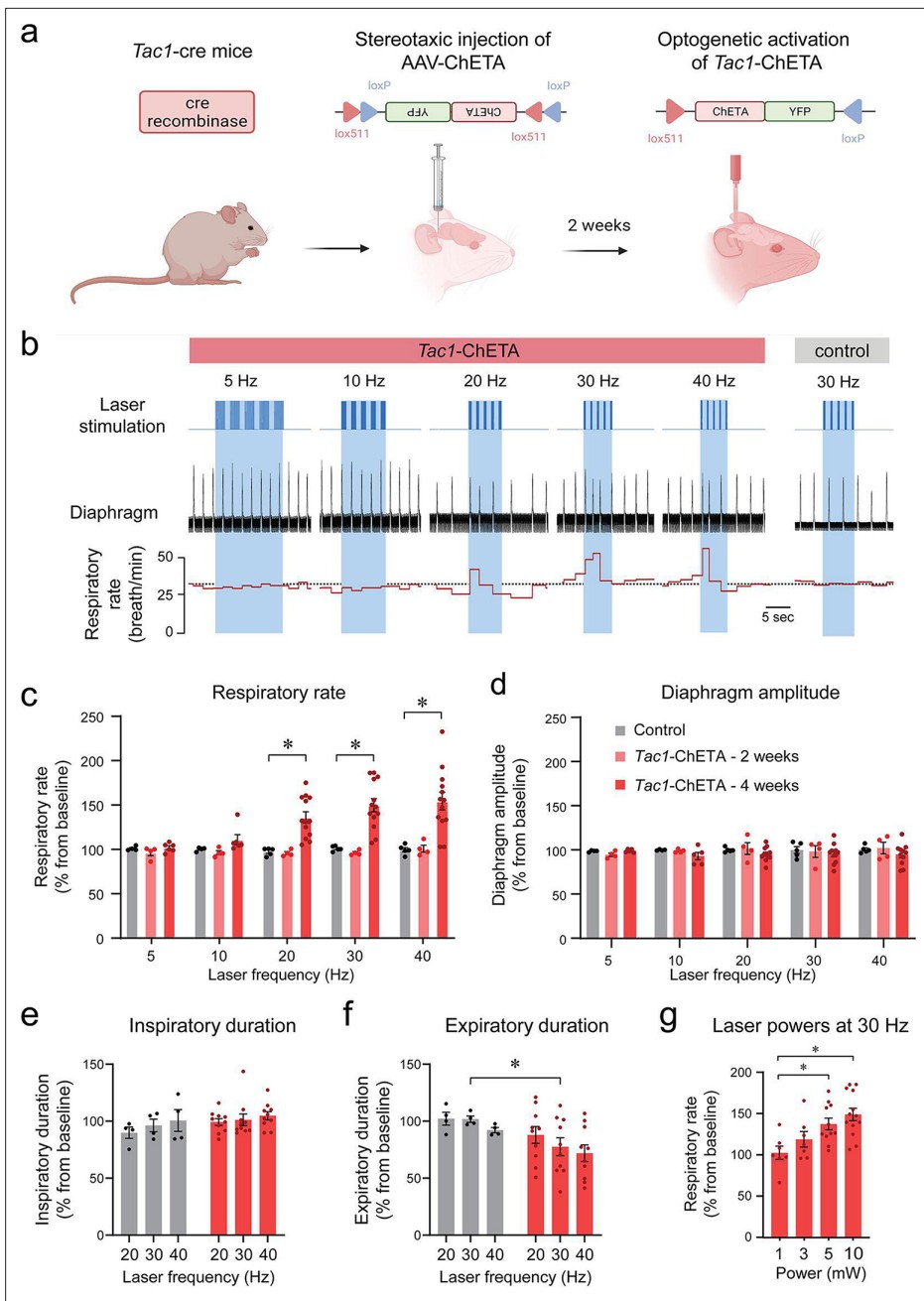

**Figure 1.** Photostimulation of tachykinin precursor 1 (*Tac1*) preBötzinger Complex (preBötC) cells increases the respiratory rate in anesthetized mice. (**a**) ChETA was expressed in *Tac1* preBötC cells by injecting the AAV-ChETA[fl/fl] virus in *Tac1* cre-expressing mice. (**b**) After 2 or 4 weeks of incubation, laser stimulations were performed at various frequencies in control and *Tac1*-ChETA anesthetized mice. (**c**) Laser stimulation increased respiratory rate at 20, 30, and 40 Hz. Increased rate following photostimulation of *Tac1* cells was only observed after 4 weeks of incubation, but not 2 weeks (n=25). (**d**) No effect was observed on diaphragm amplitude (n=25). (**e**) Laser stimulation had no effect on inspiratory duration, but (**f**) significantly decreased expiratory duration at 30 Hz (n=14). (**g**) Laser powers stimulated respiratory rate at 5 and 10 mW (n=38). Data are presented as means ± SEM, with individual data points. * indicate means significantly different from corresponding controls or laser powers with p<0.05. Panel (**a**) was created using Biorender.com.

The online version of this article includes the following source data and figure supplement(s) for figure 1:

**Source data 1.** Raw data of respiratory rate in Tac1-ChETA 2 weeks post-injection.

**Figure supplement 1.** Optogenetic stimulation of tachykinin precursor 1 (*Tac1*) preBötzinger Complex (preBötC) cells in anesthetized mice at 2- and 4 week post-injection.

**Figure supplement 1—source data 1.**

**Figure supplement 2.** Rostrocaudal distribution of the virally transduced expression of ChETA-eYFP in tachykinin precursor 1 (*Tac1*) neurons in the respiratory network.

with power of 1, 3, 5, and 10 mW significantly increased respiratory rate (*Figure 1g*; laser power effect: p=0.0018, n=38; $F_{(3,34)}$=6.201). Importantly, a minimal power of 5 mW at the tip of the optical fiber was sufficient to stimulate rhythmic breathing. According to Monte Carlo simulation (*Liu et al., 2015*), a laser power of 5 mW at the tip of the optical fibre resulted in a power density of 5 mW/mm$^2$ about 470 µm away from the optical fiber (*Figure 1—figure supplement 2*). The power density to activate 50% of ChETA channels is 5 mW/mm$^2$ (*Mattis et al., 2011*). To determine whether laser light may stimulate other respiratory neurons located outside the preBötC, we looked at the co-expression of *Tac1* and *eYFP* (for ChETA) in the medulla and determined whether it overlapped with light diffusion. Rostral to the preBötC, co-expression was found in the facial nucleus where light power was too low to activate ChETA (*Figure 1—figure supplement 2a*). The laser light was powerful enough to activate Bötzinger Complex (BötC) neurons, but co-expression was low in this region (*Figure 1—figure supplement 2b*). In the preBötC region, substantial expression was found in preBötC and the caudal ventrolateral medulla (CVLM) neurons (*Figure 1—figure supplement 2c*), but CVLM neurons are involved in cardiovascular control (*Lima et al., 2002*). Although co-expression was found caudal to the preBötC, light power was too low to activate ChETA in this region (*Figure 1—figure supplement 2d*). In summary, photostimulation of *Tac1* preBötC neurons promoted breathing by increasing respiratory rate in anesthetized mice.

## Response of *Tac1*-expressing cells depends on the stimulation phase

We produced various stimulations at different time points during the respiratory cycle with high temporal specificity using optogenetics. Frequency of 30 Hz and laser power of 10 mW were used. Stimulation phase was defined as the duration between the beginning of inspiration and the onset of the laser stimulation (*Figure 2a*). When stimulation phase was normalized to the preceding unstimulated respiratory cycle ($T_b$), simulation phase represented the percentage of the respiratory cycle when the laser was turned on, with the start of inspiration defined as 0% of the cycle and the end of expiration defined as 100% of the cycle (*Figure 2b*). All induced periods (T; duration of the induced respiratory cycle) were also normalized to the preceding unstimulated respiratory cycle ($T_b$) and presented as a percentage of $T_b$ (*Figure 2b*). When individual data points were plotted for each normalized stimulation phase from all animals, the induced period decreased when laser stimulation occurred after the first 20% of the respiratory cycle (*Figure 2b*).

Optical stimulation phases were then grouped under three categories according to their timing in the respiratory cycle: inspiration, early expiration, and late expiration (*Figure 2c*). Stimulation occurring during inspiration had no effects on the period (118% of baseline period; *Figure 2b and d*). However, stimulation decreased the period when it occurred early in expiration phase (63% of a baseline period) and also decreased the period when it occurred late in expiration phase, showing an immediate response in this phase of the respiratory cycle (85% of baseline period) (*Figure 2b and d*; stimulation phase effect: p<0.0001, n=47; $F_{(2,44)}$=18.37). Post-mortem histology confirmed that the optical fibre was positioned in regions of the medulla, dorsal to the preBötC allowing the laser light to penetrate ventral to the optical fibre (*Figure 2g*). We confirmed that ChETA expression was found in *Tac1* cells using in situ hybridization with probes for *eYFP* (ChETA, green) and *Tac1* gene (red). All cells expressing *eYFP* had a substantial expression of *Tac1* mRNAs (*Figure 2e and f* and *Figure 2—figure supplement 1*). Importantly, ChETA was expressed in a region of the medulla rich in *Tacr1* mRNAs consistent with the preBötC (*Figure 2h*). In conclusion, photostimulation of *Tac1* preBötC neurons triggered inspiration when occurring during expiration, but not during inspiration.

## *Tac1*-expressing cells: a glutamatergic subpopulation in the preBötzinger complex

As previously shown, *Tac1* cells are localized in the preBötC region (*Sun et al., 2019*). *Tac1* preBötC cells also strongly express their cognate neurokinin-1 receptors in the preBötC, and activation of neurokinin-1 receptors by substance P stimulates breathing (*Gray et al., 1999*). Knowing that the *Tac1*-expressing cells targeted in this study are mainly excitatory, we aimed to determine whether *Tac1* cells are a subpopulation of glutamatergic cells. Using in-situ hybridization, we found the presence of *Tac1*-expressing cells (red) in both the NTS and preBötC as well as the medullary raphe (*Figure 3a–d*). In the preBötC region, *Tac1* cells were expressed in 11% of cells (shown with DAPI), while glutamatergic cells (identified using the vesicular glutamate transporter VGLUT2 and associated

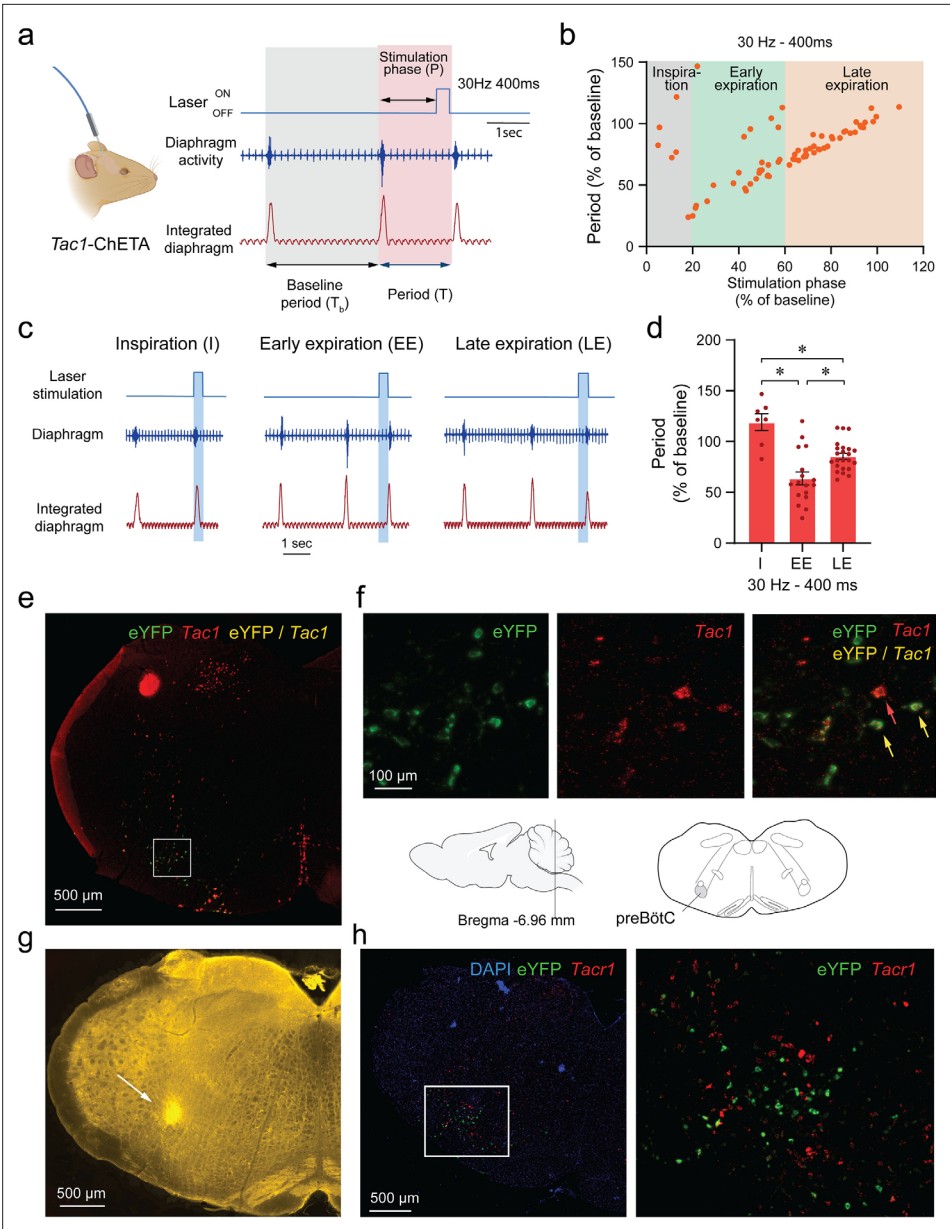

**Figure 2.** Phase-dependent photostimulation of tachykinin precursor 1 (*Tac1)* preBötzinger Complex (preBötC) cells. (**a**) For each laser stimulation, the stimulation phase (i.e. the duration between the beginning of inspiration and the onset of the laser), the induced period (**T**), and the baseline period (Tb) were measured. All stimulation phases and induced periods were normalized to the preceding unstimulated respiratory cycle (Tb) and presented as a percentage of Tb. (**b**) The changes in periods were represented against the stimulation phase at 30 Hz. When stimulation of *Tac1* cells was performed during inspiration, the period was not changed. The respiratory period was substantially reduced when stimulation occurred early in expiration and to a lesser degree late during expiration. (**c**) Laser stimulations were categorized according to their occurrences during inspiration (**I**), early (**EE**), and late expiration (**LE**). (**d**) *Tac1* stimulations occurring during early and late expirations significantly reduced the respiratory period (n=47). (**e**) Viral injection in the preBötC and opsin channel ChETA expression in *Tac1* cells was confirmed using in-situ hybridization. (**f**) *Tac1* (red) and eYFP (green) mRNA were found in the preBötC area with co-expression of both mRNAs (yellow). (**g**) The locations of the optical fiber placement above the preBötC were confirmed with post-mortem histology. Red blood cells accumulate around the optic fiber and have autofluorescence properties allowing visualization under fluorescence microscopy (*Alpert et al., 1980*). (**h**) eYFP (green) was expressed in a region rich in *Tacr1* (red) mRNA consistent with the preBötC. Data are presented as means ± SEM, with individual data points. * indicate means significantly different from stimulation phases with p<0.05. Panel a was created using Biorender.com. DAPI was shown in blue.

The online version of this article includes the following source data and figure supplement(s) for figure 2:

**Source data 1.** In-situ hybridization for Tac1 and eYFP in a DAPI stained medullary slice.

**Figure supplement 1.** In-situ hybridization showing magnified image of preBötzinger Complex (preBötC) region with co-expression of eYFP (ChETA) and tachykinin precursor 1 (*Tac1)* mRNA.

gene *Slc17a6)* were found in 44% of DAPI cells (*Figure 3e*). Interestingly, 93% of *Tac1* cells were gluta-matergic cells as shown by the co-expression of *Slc17a6* and *Tac1* mRNAs, but only 24% of *Slc17a6* cells co-expressed *Tac1* mRNAs (*Figure 3f*). These results showed that the large majority of *Tac1* cells were excitatory glutamatergic preBötC neurons.

## Stimulation of *Slc17a6*-expressing cells (*Vglut2* cells) in anesthetized mice

Knowing that *Tac1*-expressing preBötC cells are glutamatergic, we characterized the role of *Slc17a6* preBötC neurons in regulating breathing. Using a similar approach as above, we expressed the excit-atory ChETA in *Slc17a6* cells (*Figure 4a*). We produced repetitions of five stimulations with various frequencies (5, 10, 20, 30, and 40 Hz) (*Figure 4b*). Stimulations increased respiratory rate compared to the control group (*Figure 4b and c*; group effect: p=0.0019, n=11; $F_{(1,9)}$=18.77). Absolute results for respiratory rate are provided in *Figure 4—figure supplement 1*. Diaphragm respiratory amplitude was unaffected by laser stimulation (*Figure 4d*; group effect: p=0.5527, n=11; $F_{(1,9)}$=0.3803). We then determined whether inspiratory or expiratory durations were affected by laser stimulation. Inspiratory duration was not influenced by stimulation (*Figure 4e*; group effect: p=0.1476, n=10; $F_{(1,8)}$=2.569), but the expiratory duration was substantially shortened by the laser stimulation (*Figure 4f*; group effect: p=0.0057, n=10; $F_{(1,8)}$=14.03), as observed with stimulation of *Tac1* neurons. A power of 10 mW was sufficient to increase respiratory rate (*Figure 4g*; power effect: p=0.0314, n=24; $F_{(3,20)}$=3.605). To confirm viral injection in the preBötC and ChETA expression in *Slc17a6* cells, we performed in situ hybridization with probes targeting *Slc17a6* (purple) and eYFP mRNA (green). All cells tagged with eYFP were also co-expressing *Slc17a6* as shown by white cells (*Figure 4h*) in the region of the preBötC, confirming accurate expression of opsin channels in glutamatergic preBötC neurons.

## Stimulation of *Tac1* preBötC cells in freely behaving mice

To determine the role of *Tac1* preBötC cells in promoting breathing in awake animals, photostimu-lation of *Tac1* preBötC cells was performed in freely-behaving rodents while recording respiratory activity (*Figure 5a and b*). Based on the animal's behavior determined by video camera and respira-tory recordings, the behavioral states at which laser stimulation occurred, were classified according to calm and active states (*Figure 5c*). Photostimulation at 40 Hz increased respiratory rate compared to baseline during calm, but not active, state (*Figure 5d*; group effect: p=0.0003, n=27; $F_{(1.6,20.8)}$=13.59). To better quantify the changes due to photostimulation, the respiratory rate in response to photo-stimulation was expressed as a percentage of the baseline rate. During calm states, photostimulation increased respiratory rate, whereas no effect was observed during active state (*Figure 5e*; group effect: p=0.0062, n=30; $F_{(1,5)}$=20.54). During calm state, stimulation of *Tac1* preBötC cells increased respiratory rate at 30, 40, and 60 Hz, but not at 50 or 80 Hz (*Figure 5f*; group effect: p=0.0030, n=12; $F_{(1,10)}$=15.20). These effects were due to a combined decrease in both inspiratory duration (*Figure 5f*; group effect: p=0.0004, n=12; $F_{(1,10)}$=27.47) and expiratory duration (*Figure 5f*; group effect: p=0.0144, n=12; $F_{(1,10)}$=8.725). The respiratory amplitude was unaffected by photostimulation (*Figure 5—figure supplement 1*; p=0.4221, n=12; $F_{(1,10)}$=0.7008). In summary, our results demonstrated that stimulation of *Tac1* preBötC cells substantially increased respiratory rate during calm state, but not active state.

## Motor hyperactivity induced by stimulation of *Tac1*-expressing cells

To determine whether stimulation of *Tac1*-expressing cells induced behavioral changes, we first assessed locomotor activity using a video recording system and tracking software (*Figure 6a*). Repre-sentative heat maps of both control mice and mice with photostimulation of *Tac1* cells are shown in *Figure 6b* during baseline, laser stimulation, and recovery conditions. A 10 s photostimulation at 40 Hz produced significant movements in all mice (*Figure 6c*). Photostimulation increased loco-motor activity (as percentage of pixel change in the area recorded), while no change was observed in controls (*Figure 6d*; group effect: p=0.0018, n=14; $F_{(1,12)}$=15.91). Similarly, velocity was increased by photostimulation (*Figure 6e*, group effect: p=0.0163, n=14; $F_{(1,12)}$=7.792). To better assess the role of *Tac1* preBötC cells in locomotion, we measured locomotion using a rodent running wheel equipped with a sensor counting rotation speed as the number of rotations per minute (*Figure 6f*). We used a running wheel inclined at 30° allowing movements while the mouse was tethered to the laser appa-ratus. We classified stimulations according to their baseline rotation speeds and defined immobile/

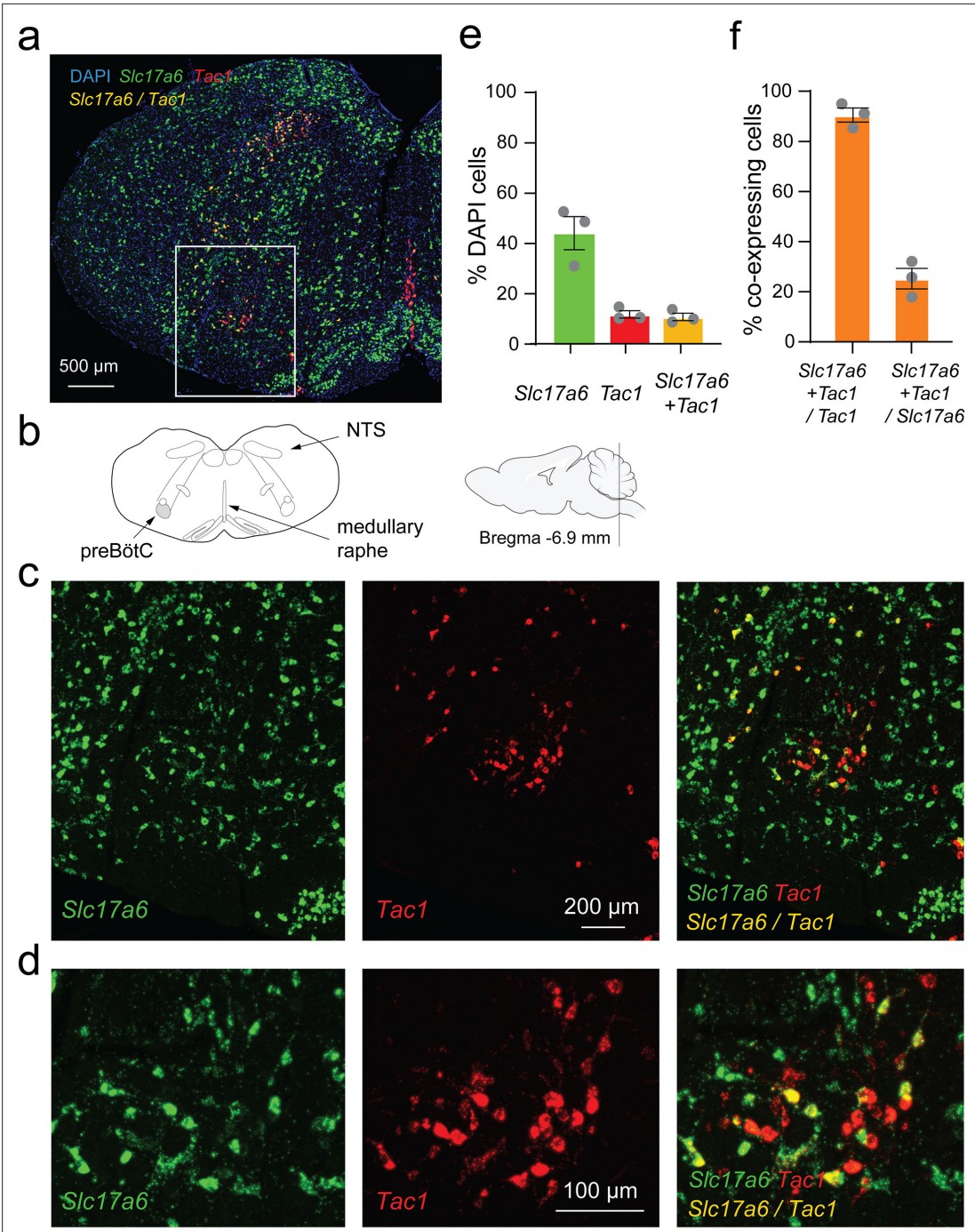

**Figure 3.** *Slc17a6* (or *Vglut2*) and tachykinin precursor 1 (*Tac1*) mRNA expressions in the medulla. (**a**) *Slc17a6* (green) and *Tac1* (red) mRNAs were expressed in the region of the preBötzinger Complex (preBötC) as shown by in-situ hybridization in wild-type mice. (**b**) Substantial expression of *Slc17a6* mRNA was observed in the medulla with a cluster of *Tac1* in the preBötzinger Complex (preBötC). (**c**) In the preBötC, part of the cells expressing *Tac1* also expressed *Slc17a6* (co-expression shown by yellow). (**d**) Magnified views of the central part of panel c. (**e**) In the preBötC, about 44% of the cells (shown with DAPI) expressed *Slc17a6*, 11% *Tac1* alone, and 11% co-expressed *Tac1* and *Slc17a6*. A total of threemice were used. (**f**) The majority of *Tac1* cells co-expressed *Slc17a6* (93%), whereas only 24% of *Slc17a6* cells co-expressed *Tac1*. DAPI was shown in blue.

The online version of this article includes the following source data for figure 3:

**Source data 1.** Raw data of respiratory rate.

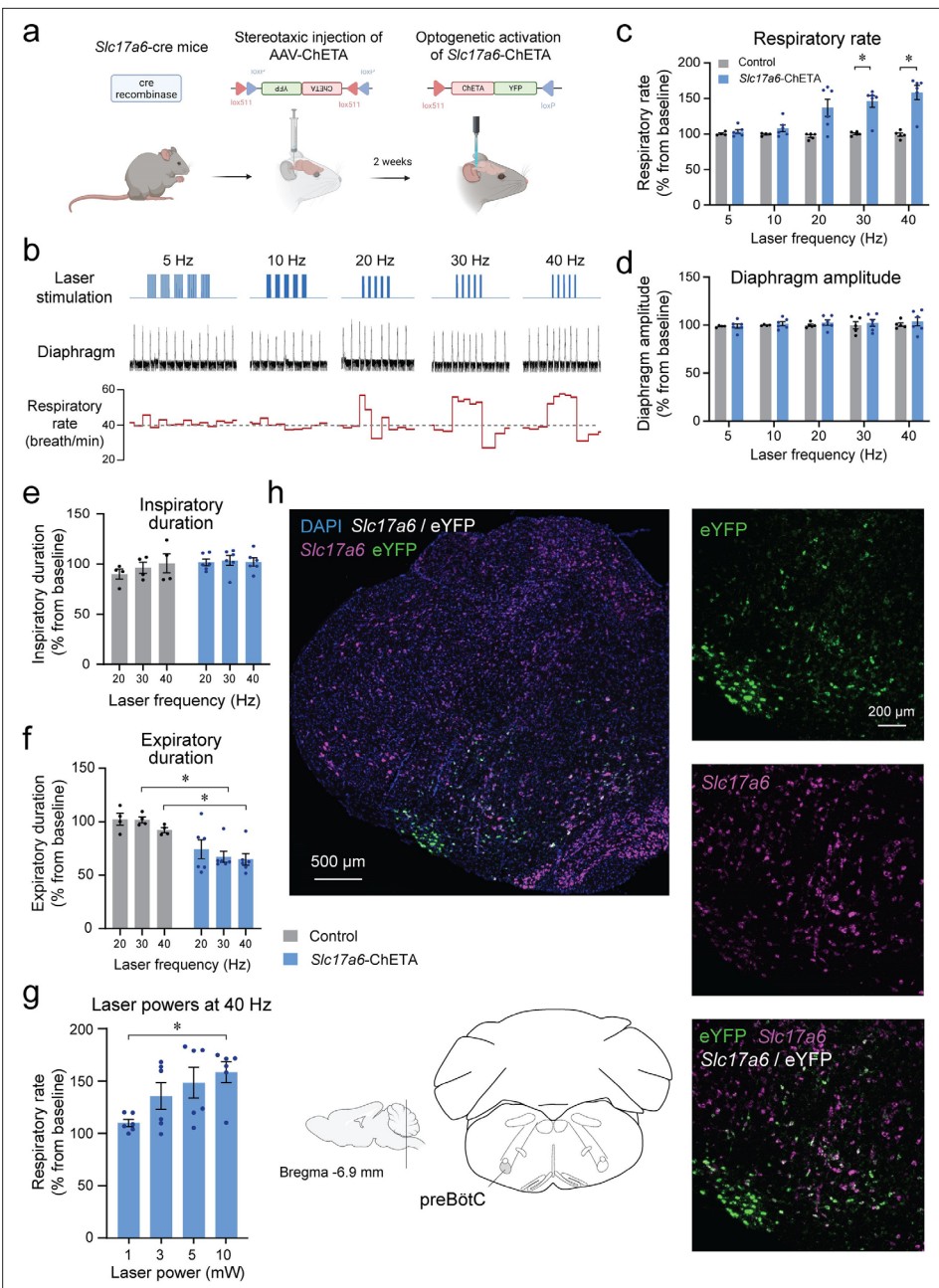

**Figure 4.** Photostimulation of *Slc17a6* (*Vglut2*) preBötzinger Complex cells increases the respiratory rate in anesthetized mice. (**a**) The adeno-associated virus AAV-ChETA^fl/fl was injected into the preBötzinger Complex (preBötC) of *Slc17a6* cre-expressing mice. After 2 weeks of incubation with AAV-ChETA, *Slc17a6* cells expressed ChETA and eYFP. (**b**) In anesthetized mice, diaphragm muscle activity was recorded and laser stimulations at various frequencies were performed in the preBötC. (**c**) Respiratory rate was significantly increased by laser stimulation at 30 and 40 Hz and (**d**) no effect was observed on diaphragm amplitude (n=11). Increased respiratory rate was not due to a decrease in (**e**) inspiratory duration, but rather decreased (**f**) expiratory duration (n=11). (**g**) Respiratory rate was increased by laser stimulation at 10 mW (n=24). (**h**) In situ hybridization was performed in animals injected with AAV-ChETA and showed that ChETA (marked with eYFP mRNA) was expressed in the region of the preBötC and co-expressed with *Slc17a6*. Data are presented as means ± SEM, with individual data points. * indicate means significantly different from corresponding controls or laser power with p<0.05. Panel (**a**) was created using Biorender.com.

The online version of this article includes the following source data and figure supplement(s) for figure 4:

**Source data 1.** Respiratory rate in response to stimulation of Slc17a6-ChETA in mice.

**Figure supplement 1.** Optogenetic stimulation of *Slc17a6* preBötzinger Complex (preBötC) cells in anesthetized animals.

**Figure supplement 1—source data 1.**

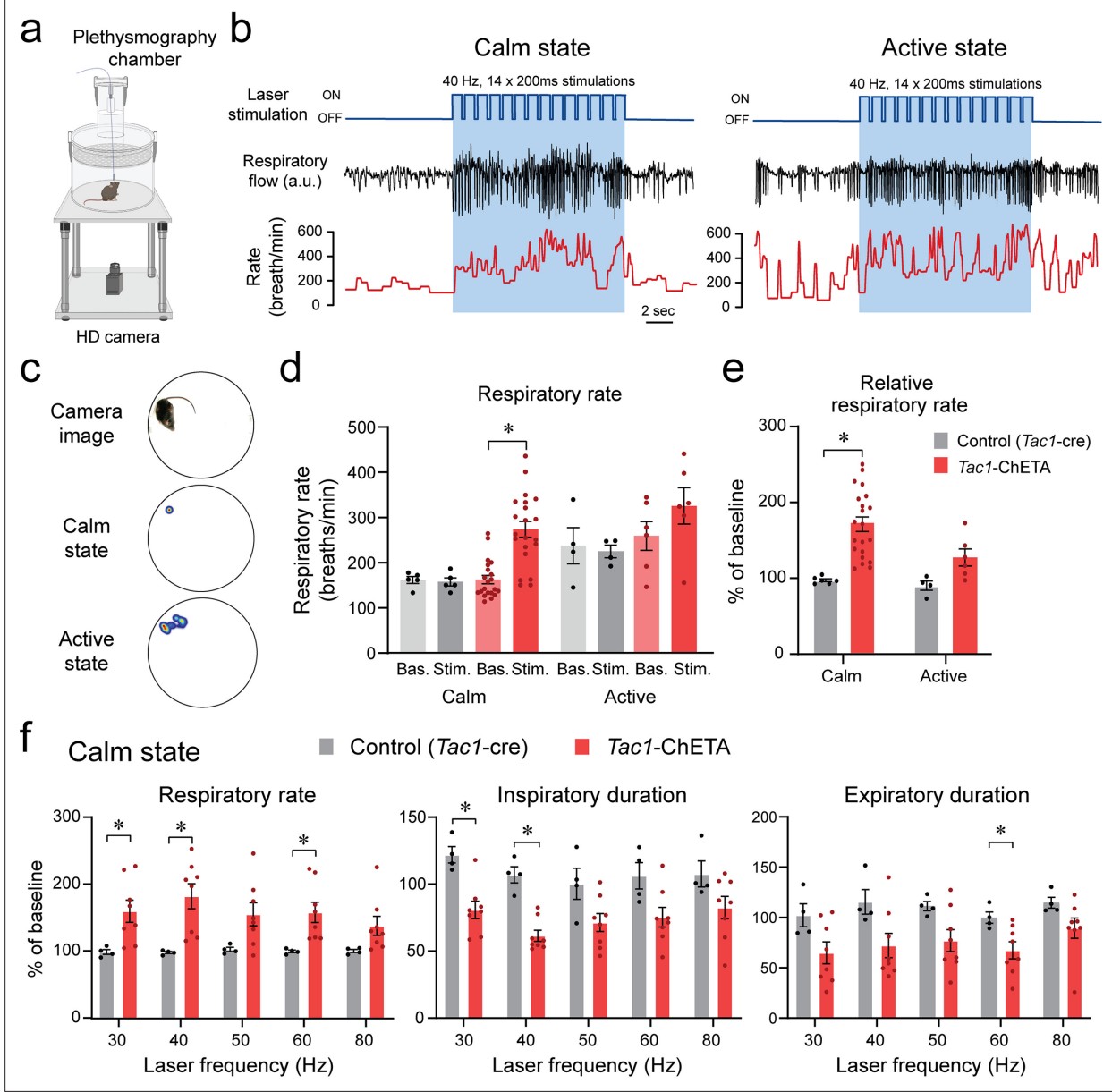

**Figure 5.** State-dependent respiratory changes by photostimulation of tachykinin precursor 1 (*Tac1*) preBötzinger Complex (preBötC) cells in freely-behaving mice. (**a**) Using viral injection, ChETA was expressed in *Tac1* preBötC cells and respiratory activity was measured using whole-body plethysmography. (**b**) Photostimulation of ChETA at 40 Hz increased respiratory rate in calm, but not in active state, with this effect reversed when the laser was turned off. (**c**) According to video recordings, calm state was determined when the animal was not moving, whereas the active state was when activity was higher than 0.4% of pixel changes in the recording area. (**d**) Stimulation of *Tac1* neurons strongly increased absolute respiratory rate in calm state whereas in active states it had no significant effect. (**e**) To determine the state-dependent effects of photostimulation, respiratory rate changes were expressed as a function of the baseline respiratory rate (before laser stimulation). Photostimulation of *Tac1* cells substantially increased respiratory rate in calm, but not in active, state. (**f**) In calm state, *Tac1* cell stimulations at 30, 40, and 60 Hz increased rate due to a combination of decreased inspiratory and expiratory durations (n=12). Data are presented as means ± SEM, with individual data points. * indicate means significantly different from corresponding controls with p<0.05. Panel (**a**) was created using Biorender.com.

The online version of this article includes the following source data and figure supplement(s) for figure 5:

**Source data 1.** Raw data for expiratory duration in response to photostimulation.

**Figure supplement 1.** Tidal volume following optogenetic stimulation of tachykinin precursor 1 (*Tac1*) preBötzinger Complex (preBötC) cells in freely-behaving mice.

**Figure supplement 1—source data 1.** Tidal volume in response to Tac1 stimulation.

low running when rotation was <20 rotations/min and running when speed was ≥20 rotations/min. When immobile or slow running, photostimulation of *Tac1* preBötC cells substantially increased rotation speed (*Figure 6g and h*, state x stimulation effect: p=0.0015, n=14; $F_{(1,12)}$=16.64). However, when initially running, photostimulation only moderately increased rotation speed (*Figure 6g and h*). In summary, we showed that photostimulation of *Tac1* cells substantially increased locomotion using two different assays in freely-behaving mice.

Considering that the locomotor effects of *Tac1* preBötC stimulation may be due to stimulation of *Tac1* neurons in nearby motor nuclei, we performed in-situ hybridization for *Tac1* and eYFP-ChETA mRNAs in distinct regions of the medulla. Low co-expression of *Tac1* and eYFP/ChETA was found in the BötC located rostral to the preBötC (*Figure 6—figure supplement 1a*). Substantial co-expression was found in the preBötC, moderate co-expression in the caudoventrolateral reticular nucleus (CVLM), and low co-expression in the lateral paragigantocellular nucleus (LPGi) and the ventral spinocerebellar tract (VSC) (*Figure 6—figure supplement 1b*). Caudal to the preBötC, low co-expression was found in the lateral reticular nucleus (LRt) (*Figure 6—figure supplement 1c*). Considering that light power decreased as it diffused away from the optical fiber, only the BötC and LPGi neurons were likely to be activated by light. Although the LPGi modulates motor control (*Capelli et al., 2017*), only a few *Tac1*-positive neurons were found in this region which is unlikely to contribute to the locomotor effects of *Tac1* cell stimulation. Overall, using two different locomotor assays, we showed that photostimulation of *Tac1* neurons promoted locomotor activity, likely by activating neurons in the preBötC.

## Co-expression of *Tac1* and *Oprm1* in the preBötzinger complex

Considering the role of *Tac1*-expressing cells in rhythmic breathing and knowing that the preBötC plays a major part in respiratory depression by opioid drugs (*Montandon et al., 2011*; *Stucke et al., 2015*), we determined whether *Tac1* mRNA is co-expressed with *Oprm1* (gene for MORs) mRNA in the preBötC. We performed in situ hybridization in medullary sections containing the preBötC in wild-type mice (C57BL/6 J; *Figure 7a and b*). In the region of the preBötC, *Oprm1* was expressed in 40.5 ± 6.8% of DAPI-stained cells while *Tac1* was expressed in 21.8 ± 8.9% of DAPI-stained cells (*Figure 7c*; n=3). Co-expression of both *Oprm1* and *Tac1* was found in 17.7 ± 7.6% of DAPI-stained cells (*Figure 7c*; n=3). Interestingly, most cells expressing *Tac1* also co-expressed *Oprm1* (75.9 ± 5.7% of *Tac1*-expressing cells; *Figure 7c and d*; n=3) and a smaller fraction of *Oprm1* cells were expressing *Tac1* (38.3 ± 13.7% of *Oprm1*-expressing cells; *Figure 7c and d*; n=3). In summary, *Tac1* cells constituted a small fraction of cells found in the preBötC but a large proportion of these cells co-expressed *Oprm1* mRNA.

## Stimulation of *Tac1*-expressing cells reverses opioid-induced respiratory depression

Considering that photostimulation of *Tac1* cells increased breathing and that these cells co-expressed *Oprm1*, we aimed to determine whether stimulation of *Tac1*-expressing cells can reverse opioid-induced respiratory depression. We injected the opioid fentanyl (intramuscular; 5 µg/kg) in anesthetized mice followed by stimulations at 20, 30, and 40 Hz (*Figure 8a and b*). As expected, fentanyl injection reduced respiratory rate to 53.6% of baseline value, and laser stimulations with frequencies at 20, 30, and 40 Hz significantly reversed fentanyl-induced respiratory rate depression (*Figure 8c and d*; stimulation effect: p=0.0002, n=35; $F_{(4,30)}$=7.986, *Figure 8—figure supplement 1*). To determine the effect of photostimulation on respiratory depression by opioid drugs in more realistic conditions and without anesthetics, we performed similar experiments in freely-behaving rodents. We used whole-body plethysmography to assess respiratory responses to the opioid fentanyl (300 µg/kg, intraperitoneal) (*Figure 8e*). To consider the stress associated with intraperitoneal injection in live mice, saline injections were used as controls. In mice injected with saline, respiratory rate increased following injection due to mouse handling and stress, and photostimulation at 40 Hz did not significantly change respiratory rate. Injection of fentanyl did not increase respiratory rate, and showed a significantly lower respiratory rate compared to mice injected with saline. Photostimulation significantly increased respiratory rate (*Figure 8f and g*; group x stimulation effect: p<0.0001, n=14; $F_{(2,24)}$=16.84), despite fentanyl lowering respiratory rate compared to saline. Interestingly, the saline injection did not change the tidal volume while fentanyl injection increased it. Photostimulation did not change tidal volume (*Figure 8f and g*; group x stimulation effect: p<0.0001, n=14; $F_{(2,24)}$=16.42). Mouse movements were

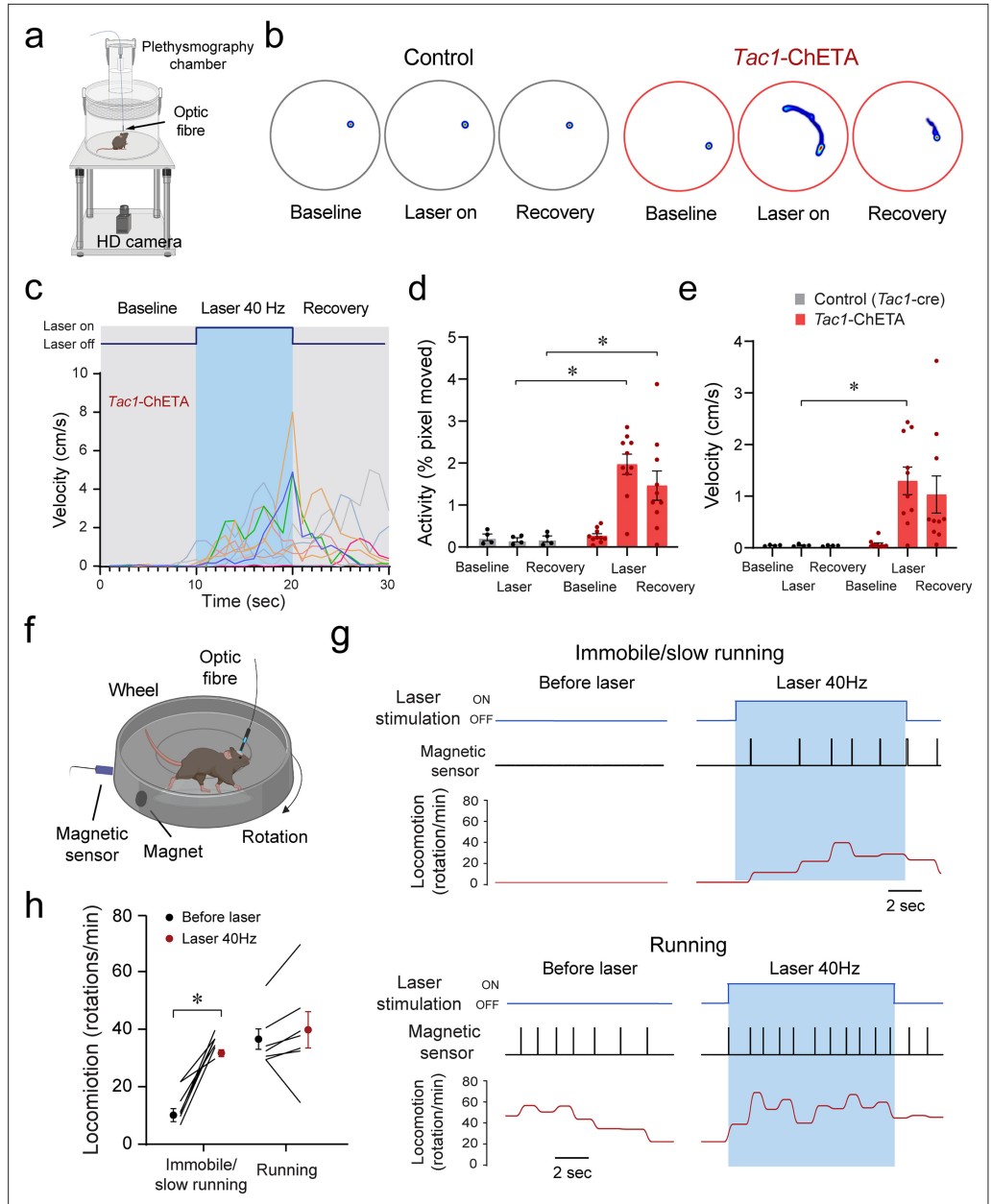

**Figure 6.** Optogenetic stimulation of tachykinin precursor 1 (*Tac1*) preBötzinger Complex (preBötC) cells promotes locomotion in freely-behaving mice. (**a**) Locomotor activity in freely behaving *Tac1*-cre mice expressing ChETA was assessed with a high-definition camera. (**b**) Heat maps showing locomotion in control and *Tac1*-ChETA mice in three different conditions; before stimulation (baseline), during stimulation (laser ON), and following stimulation (recovery). Each circle represents 10 s locomotion under the corresponding condition. (**c**) The velocity (cm/s) for each *Tac1*-ChETA animal strongly increased during stimulation at 40 Hz followed by a reduction in velocity after photostimulation was stopped. (**d**) Activity (pixel changes inside the recording circle) also significantly increased with laser stimulation in *Tac1*-ChETA, but not control, mice (n=14). (**e**) Similarly, photostimulation substantially increased velocity in *Tac1*-ChETA compared to control mice, with this effect sustained for a few seconds during recovery (n=14). (**f**) Using a running wheel, locomotion was measured as rotation speed while *Tac1* preBötC cells were stimulated. (**g**) When the animal was immobile with a rotation speed inferior to 20 rotations/min, photostimulation of *Tac1* preBötC cells increased substantially rotation speed. When the animal was already running, photostimulation did not change rotation speed. (**h**) Photostimulation of *Tac1* neurons significantly increased rotation speed in immobile mice, but not in running mice (n=14). Data are presented as means ± SEM, with individual data points. * indicate means significantly different from corresponding controls with p<0.05. Panels (**a**) and (**f**) were created using Biorender.com.

The online version of this article includes the following source data and figure supplement(s) for figure 6:

**Source data 1.** Locomotion in response to Tac1 photostimulation.

**Figure supplement 1.** mRNA expressions of eYFP (for ChETA) and tachykinin precursor 1 (*Tac1*) (for substance P) in the preBötzinger Complex region and adjacent motor nuclei.

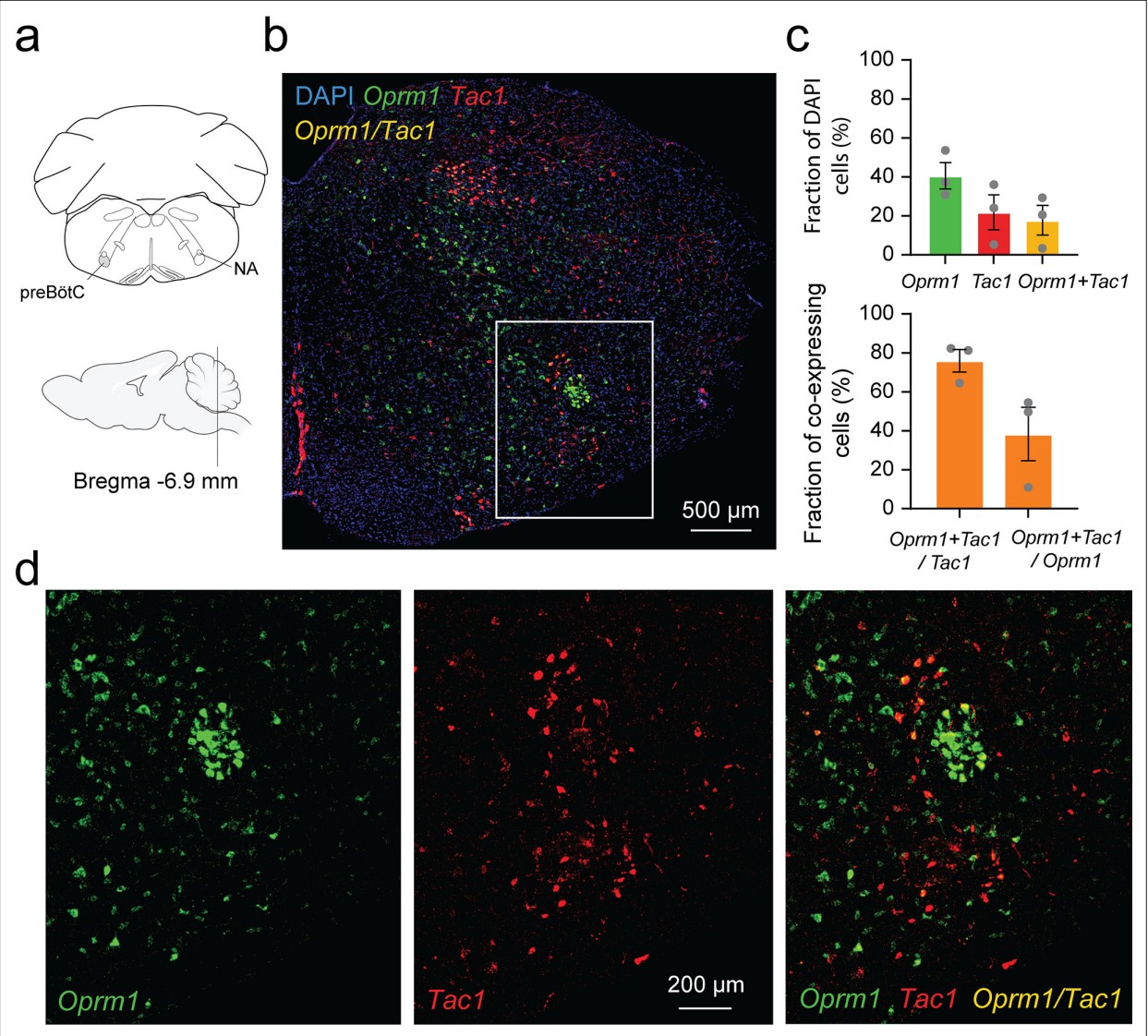

**Figure 7.** Co-expression of *Oprm1* and tachykinin precursor 1 (*Tac1*) mRNAs in preBötzinger Complex (preBötC) cells. (**a**). In-situ hybridization was performed on sections containing the preBötC about 6.9 mm caudal to Bregma. (**b**) In the medulla, *Oprm1* (the gene encoding for MOR shown in green), *Tac1* (the gene for Substance P in red), and DAPI (in blue) were widely expressed in the preBötC. (**c**) Cell counting in the preBötC shows that about 40.5% of the cells contained *Oprm1* and 21.8% contained *Tac1* (n=3). In the preBötC, 75.9% of cells expressing *Tac1* also co-expressed *Oprm1* (n=3). Conversely, about 38% of cells expressing *Oprm1* also co-expressed *Tac1* (n=3). (**d**) In a magnified view of the preBötC, *Oprm1* mRNAs formed a cell cluster in the ventral part of the medulla and co-expressed *Tac1*. Na: nucleus ambiguus.

The online version of this article includes the following source data for figure 7:

**Source data 1.** Expression of Tac1 and Oprm1 mRNAs.

then analyzed following saline or fentanyl injection. Fentanyl injections increased velocity in mice and laser stimulation further raised it (***Figure 8h and i***; group x stimulation effect: p=0.0941, n=13; $F_{(2,22)}$=2.637). In conclusion, fentanyl presented a lower respiratory rate compared to saline injection, which was fully reversed by stimulation of *Tac1* preBötC neurons in anesthetized and freely-behaving mice.

## Discussion

Characterizing the neuronal elements of the preBötzinger Complex is key to understanding the complexity of the network generating breathing and to identify therapeutic targets when breathing

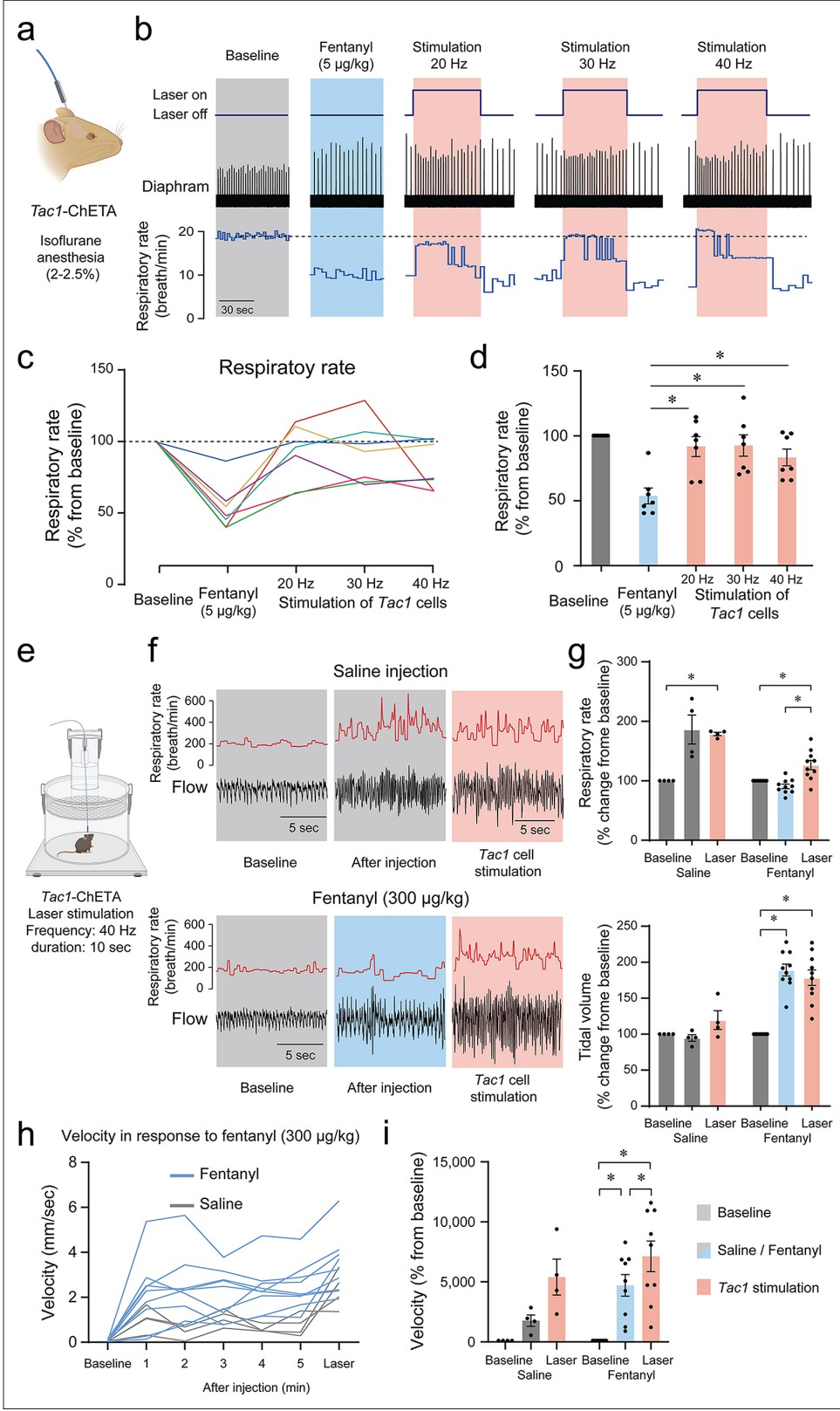

**Figure 8.** Photostimulation of tachykinin precursor 1 (*Tac1*) preBötzinger Complex (preBötC) cells reverses respiratory depression by the opioid fentanyl. (**a**) Injection of fentanyl (5 µg/kg) performed in anesthetized *Tac1*-ChETA mice. (**b**) Injection of fentanyl depressed breathing, with this effect reversed by photostimulation of *Tac1* preBötC cells at 20, 30, and 40 Hz. (**c**) All seven animals receiving fentanyl showed respiratory rate depression by fentanyl reversed by *Tac1* photostimulation (each color represents a separate tracing for each animal). Once the laser was turned off, the breathing rate returned

*Figure 8 continued on next page*

Figure 8 continued

to low breathing rates due to fentanyl. (**d**) Mean data showed that stimulation of *Tac1* cells reversed respiratory depression at each laser frequency used (20, 30, 40 Hz, n=35). (**e**) In freely-moving, non-anesthetized, *Tac1*-ChETA mice, respiratory responses to the opioid fentanyl (300 µg/kg, intraperitoneal) were assessed using whole-body plethysmography. (**f**) Representative tracings of diaphragm activity and respiratory rate showed that fentanyl depressed respiratory rate compared to saline and that *Tac1* stimulation reversed respiratory depression. (**g**) Mean data show how fentanyl significantly reduced respiratory rate observed with saline injection, an effect reversed by *Tac1* stimulation in fentanyl conditions only (n=14). Tidal volume was increased by fentanyl injection but unaffected by photostimulation. (**h, i**) Locomotor activity was assessed and showed an increase in velocity following injection and stimulation at 40 Hz (n=13). All absolute values (not normalized according to baseline) of respiratory rate and tidal volume can be found in *Figure 8—figure supplement 1* and are consistent with normalized results. Data are presented as means ± SEM, with individual data points. * indicate means significantly different from corresponding controls with p<0.05. Panel (**a**) and (**e**) were created using Biorender.com.

The online version of this article includes the following source data and figure supplement(s) for figure 8:

**Source data 1.** Raw data for respiratory rate in response to fentanyl and photostimulation of Tac1 cells.

**Figure supplement 1.** Fentanyl depression and optogenetic stimulation of tachykinin precursor 1 (*Tac1*) preBötzinger Complex (preBötC) cells.

**Figure supplement 1—source data 1.** Tidal volume in animals receiving saline or fentanyl.

fails. We characterized a subpopulation of glutamatergic neurons expressing the *Tac1* gene promoting breathing that can be targeted to reverse respiratory depression by opioid drugs. By unilaterally modulating *Tac1* preBötC cells, we modulated rhythmic breathing in a spatially and temporally precise manner in both anesthetized and freely moving mice. This is the first in vivo evidence that optogenetic activation of *Tac1* preBötC cells increased breathing, reset the inspiratory cycle in a phase-dependent manner, and induced a strong behavioral response. The role of *Tac1* cells iand gastrointestinal regulation and gastrointestinal regulation n breathing depended on the state of the animal (calm *vs.* active) and, therefore, the overall excitability of the respiratory network. A minimal effect of *Tac1* cell stimulation was observed when the animal was behaviorally active, whereas substantial increases were observed when the animal was calm.

## *Tac1*-expressing cells, a small subpopulation of glutamatergic cells, regulate breathing

The preBötzinger complex is a heterogeneous network containing multiple subpopulations of glutamatergic and GABA/glycinergic neurons balancing excitation and inhibition to produce rhythmic breathing (*Del Negro et al., 2018*). In the preBötC, most inspiratory-related excitatory cells are glutamatergic, and their activation is essential to initiate synchronized rhythmic activity and produce a breath (*Wallén-Mackenzie et al., 2006*). Stimulation of glutamatergic excitatory cells in the preBötC entrains respiratory activity both in vitro and in vivo (*Baertsch et al., 2018*; *Oliveira et al., 2021*). However, much remains unknown about the different subclasses of glutamatergic preBötC neurons and how they produce rhythmic breathing. A large subset of excitatory preBötC neurons derive from the transcription factor developing brain homeobox 1 protein (Dbx1) and are potential drivers of rhythmic breathing (*Baertsch et al., 2018*; *Vann et al., 2018*). In the present study, we focused on a limited group of excitatory neurons expressing the neuropeptide substance P (encoded by the *Tac1* gene). Substance P is widely spread in the central and peripheral nervous system including the brainstem and the preBötC. It is involved in respiratory, cardiovascular, and gastrointestinal regulation as well as nociception and chemoreception but the function of *Tac1*-expressing cells in the regulation of breathing in vivo is unknown (*Otsuka and Yoshioka, 1993*; *Gray et al., 1999*; *Hökfelt et al., 2001*; *Peña and Ramirez, 2004*; *Ptak et al., 2009*). In vitro, substance P increases respiratory rhythm by activating neurokinin-1 receptors (*Gray et al., 1999*; *Liu et al., 2004*; *Peña and Ramirez, 2004*; *Yeh et al., 2017*; *Sun et al., 2019*). Neurokinin-1 receptors are G-protein-coupled receptors stimulating rhythmic breathing through various ion channels including GIRK channels (*Montandon et al., 2016a*; *Yeh et al., 2017*). *Tacr1*-expressing preBötC cells are predominantly glutamatergic and mediate inspiratory activity (*Guyenet et al., 2002*). Interestingly, the destruction of neurokinin-1 receptor-expressing preBötC cells using saporin-Substance P progressively disrupts breathing and eventually stops it (*Gray et al., 2001*), with this effect more pronounced during sleep (*McKay and Feldman, 2008*). Substance P expression is also consistent with the immunohistochemical distribution of Dbx1-derived neurons in the preBötC (*Sun et al., 2019*). In the present study, activation of *Tac1*-expressing preBötC cells promoted inspiration to similar levels than photostimulation of glutamatergic preBötC.

According to in situ hybridization, *Tac1*-expressing cells constituted less than 24% of glutamatergic preBötC cells. Most cells expressing eYFP-ChETA were found ventral to the nucleus ambiguus and were not found in other regions involved in the generation of rhythmic breathing such as the post-inspiratory complex (PiCo) (*Figure 1—figure supplement 1*). However, cells expressing eYFP-ChETA were found ventral and rostral to the preBötC in regions containing C1 adrenergic cells which are known to modulate respiratory rate and arousal (*Burke et al., 2014*). Considering the role of C1 cells in active expiration (*Malheiros-Lima et al., 2018*) combined with the strong inspiratory activity observed following photostimulation, our study strongly suggests that *Tac1*-expressing preBötC cells constitute a population of glutamatergic preBötC cells sufficient to generate inspiratory activity in vivo.

## Phase-dependent role of *Tac1*-expressing cells

Photostimulation of *Tac1*-expressing preBötC cells produced inspiratory activity only during discrete periods of the respiratory cycle. *Tac1* photostimulation had no effect during the first 20% of the respiratory cycle but triggered inspiratory activity when applied during post-inspiration or expiration. These results are consistent with the lack of effects when *Slc17a6*-expressing (glutamatergic) or Dbx1-derived cells were stimulated during inspiration in anesthetized mice (*Oliveira et al., 2021*). Consistent with our results, stimulation of *Slc17a6*-expressing, but not Dbx1-derived, cells produced inspiration when photostimulation was applied during expiration (*Oliveira et al., 2021*). On the other hand, stimulation of Dbx1-derived neurons produced inspiration 2 s after endogenous inspiratory bursts in brainstem slides in vitro (*Kottick and Del Negro, 2015*). A smaller population of excitatory SST-expressing cells (which are 87% glutamatergic) also showed excitatory effects, where it prolonged the respiratory period in anesthetized mice but shortened the period when stimulation was performed during mid-expiration (*de Sousa Abreu et al., 2022*). In this study, the early refractory phase during which *Tac1* preBötC cells cannot initiate inspiration implies that *Tac1* cells may either be temporarily inhibited by glycinergic and/or GABAergic neurons (*Sherman et al., 2015*) or may present reduced network excitability after an inspiratory burst (*Baertsch et al., 2018*) due to prolonged afterhyperpolarization or depletion of presynaptic vesicles (*Baertsch et al., 2018*).

## *Tac1* preBötC cells and motor behaviors

The preBötC is the site of respiratory rhythmogenesis and is critical to generate inspiration in mammals. The preBötC is not limited to its main role in the generation of breathing. A subset of preBötC cells derived from Dbx1 and expressing cadherin are also involved in arousal, and their ablation increased slow-wake cortical activity (*Yackle et al., 2017*), suggesting that the preBötC may play roles beyond rhythmic breathing. Outside of this region, activation of *Tac1* neurons in the preoptic area of the hypothalamus promoted arousal and enhanced locomotor activity (*Reitz et al., 2021*). Substance P is also a key-neuropeptide involved in nociception (*Mantyh, 2002*) and locomotion (*Farrell et al., 2021*). Descending *Tac1* brainstem circuits mediate behavioral responses, such as the fight-or-flight response, associated with brisk locomotor activity (*Barik et al., 2018*; *Kuwaki, 2021*). To anticipate the body's metabolic demand in the event of locomotor nocifensive response, nociceptive stimuli elicit cardio-respiratory responses, such as augmented breathing (*Jafari et al., 2017*) and cardiovascular response in conscious rodents (*Unger et al., 1988*) In our study, stimulation of *Tac1* cells promoted locomotion. It is plausible that this locomotor response was due to photostimulation of adjacent motor nuclei. ChETA-eYFP expression was observed in the caudal ventrolateral reticular nucleus which is involved in nociceptive-cardiovascular integration (*Lima et al., 2002*) suggesting that this nucleus may not contribute directly to the increased locomotor activity. The Ventral SpinoCerebellar tract and the Lateral ParaGigantocellular nucleus are both involved in locomotor control (*Capelli et al., 2017*; *Chalif et al., 2022*), but their low ChETA-eYFP expression and their relative distance from the optical fiber suggest that these regions did not mediate the locomotive effects observed with photostimulation. Caudal to the preBötC is the dorsal part of the medullary reticular nucleus and the lateral reticular nucleus which control escape responses to noxious stimuli and contain *Tac1*-expressing cells (*Barik et al., 2018*). However, both regions show very low eYFP-ChETA and are not within the range of light diffusion. In conclusion, stimulation of *Tac1* neurons induced a strong locomotor response that is likely due to photostimulation of preBötC cells, but it cannot be fully excluded that some motor nuclei may also partially contribute to this response.

## Reversal of respiratory depression by opioid drugs

Opioid drugs present unwanted side effects such as respiratory depression. The respiratory properties of opioid drugs are due to their action on µ-opioid receptors (*Heinricher et al., 2009*; *Montandon, 2022a*). The preBötC expresses MORs and mediates a large component of respiratory rate depression by opioid drugs (*Montandon et al., 2011*; *Bachmutsky et al., 2020*; *Varga et al., 2020*). Interestingly, preBötC neurons expressing neurokinin-1 receptors, the cognate receptors of substance P, are preferentially inhibited by opioid drugs (*Montandon et al., 2011*). In our study, we found co-expression of *Oprm1* (the gene coding for MORs) and *Tac1* in preBötC neurons, and stimulation of *Tac1* preBötC cells entirely reversed respiratory depression by fentanyl. Consistent with these results, opioid drugs directly inhibit substance P-expressing cells and substance P production in the spinal cord (*Fukazawa et al., 2007*). Moreover, an excitatory *Tac1* spinal-brainstem circuit mediates noxious responses in rodents (*Barik et al., 2018*; *Gutierrez et al., 2019*) and deletion of the *Tac1* gene in mice increases the inhibitory effects of opioid drugs, suggesting that substance P could reverse respiratory depression by opioid drugs (*Takita et al., 2000*; *Berner et al., 2012*). The tachykinin system (substance P release and activation of neurokinin-1 receptors) may, therefore, constitute a tonic excitatory circuit that could antagonize the suppressive effects of opioids. Consistent with this hypothesis, we showed that stimulation of *Tac1* preBötC cells alleviated respiratory depression by fentanyl in intact animals. Interestingly, stimulation of breathing by neurokinin-1 receptor activation in the preBötC is regulated by GIRK (*Montandon et al., 2016a*), which also regulates respiratory rate depression by opioid drugs (*Montandon et al., 2016b*). Such converging cellular mechanisms further support the idea of a link between *Tac1* neurons and preBötC inhibition due to MORs.

The brainstem circuits generating and regulating rhythmic breathing are complex. At the core of the respiratory circuits is the preBötC, a collection of neurons with diverse neurochemical identities. Here, we identified a sub-population of glutamatergic neurons, expressing the precursor *Tac1* of the neuropeptide substance P, that can trigger inspiration and can raise breathing in freely-behaving mice. The importance of *Tac1* preBötC neurons is highlighted by the fact that when breathing is inhibited by opioid drugs or anesthetics, activation of *Tac1* preBötC neurons is sufficient to generate inspiration. Interestingly, stimulation of *Tac1* neurons also triggers locomotion, therefore, suggesting that *Tac1* preBötC neurons may play roles beyond breathing.

# Materials and methods
## Key resources table

| Reagent type (species) or resource | Designation | Source or reference | Identifiers | Additional information |
|---|---|---|---|---|
| Strain, strain background (*M. musculus*) | C57BL/6 J | The Jackson Laboratory | # 000664 | |
| Genetic reagent (*M. musculus*) | *Slc17a6*-ires-cre *Slc17a6*tm2(cre)Lowl/J | The Jackson Laboratory | # 016963 | |
| Genetic reagent (*M. musculus*) | *Tac1*-IRES2-Cre-D B6;129S-*Tac1*tm1.1(cre)Hze/J | The Jackson Laboratory | # 021877 | |
| Biological sample (*M. musculus*) | Mouse isolated brainstem | St. Michael's hospital/ Animal care facility | | Isolated from mouse and fixed in formalin |
| Recombinant DNA reagent | AAV$_5$-EF1a-DIO-ChETA-eYFP | UNC vector core | AV4322c | |
| Recombinant DNA reagent | AAV-EF1a-DIO-eYFP-WPRE-pA | UNC vector core | AV4310L | |
| Commercial assay or kit | RNAscope Multiplex Fluorescent Reagent v2 Assay | Advanced Cell Diagnostics | # 323270 | |
| Commercial assay or kit | Mm-*Tac1* | Advanced Cell Diagnostics | Cat No. 410351-C2 | Probe targeting *Tac1* gene mRNA |
| Commercial assay or kit | Mm-*Slc17a6* | Advanced Cell Diagnostics | Cat No. 319171-C2 | Probe targeting *Slc17a6* gene mRNA |

*Continued on next page*

*Continued*

| Reagent type (species) or resource | Designation | Source or reference | Identifiers | Additional information |
|---|---|---|---|---|
| Commercial assay or kit | Mm-*Oprm1* | Advanced Cell Diagnostics | Cat No. 315841 | Probe targeting *Oprm1* gene mRNA |
| Commercial assay or kit | Mm-*Tacr1* | Advanced Cell Diagnostics | Cat No. 428781 | Probe targeting NK1 receptor gene mRNA |
| Commercial assay or kit | Mm-eYFP | Advanced Cell Diagnostics | Cat No. 551621-C3 | Probe targeting eYFP gene mRNA |
| Chemical compound, drug | Fentanyl citrate | Sandoz | St. Michael's Hospital Pharmacy | |
| Software, algorithm | Labchart | ADInstruments | | Version 8 |
| Software, algorithm | GraphPad Prism 9 | Graphpad Software | | Version 9.3.1 |
| Software, algorithm | Adobe Illustrator | Creative Suite 5, Adobe | | |
| Software, algorithm | OptogenSim | *Liu et al., 2015* | DOI:10.1364/BOE.6.004859 | 3D Monte Carlo simulation platform for optogenetic applications |
| Software, algorithm | Fiji (ImageJ) | *Schindelin et al., 2012* | doi:10.1038/nmeth.2019 | |
| Software, algorithm | Zen 2.6 Lite | Zeiss | | (Blue edition) |
| Software, algorithm | Pinnacle Studio 24 MultiCam Capture software | Corel | | |
| Software, algorithm | EthoVision XT | Noldus | | Version 14 |
| Other | DAPI | Biotium | #40011 | 4',6-diamidino-2-phenylindole (fluorescent marker binding to DNA) |
| Other | Needle | P1 technologies | C315I-SPC Internal 33GA | Virus injection in brain |
| Other | custom-made ferule/ optical fiber | Thor Labs | Ferule: CF230-10 Optical fiber: FP200ERT | Specifications: 200 µm, 0.5 NA |
| Other | Optical fiber | Thor Labs | CFMC52L10 | Specifications: 200 µm, 0.5 NA |

## Animal care, drug, and virus acquisition

All procedures were carried out in accordance with the recommendations of the Canadian Council on Animal Care and were approved by St. Michael's Hospital animal care committee (animal use protocols #981 and #988). Experiments were performed on 63 adult mice of either sex, aged between 3–4 months old and weighing between 20 g and 40 g. Animals were kept on a 12 hr light–dark cycle with unrestricted food and water and all experiments were performed during the day. Slc17a6-ires-cre (STOCK *Slc17a6*tm2(cre)Lowl/J; Strain # 016963) and *Tac1*-IRES2-Cre-D (B6;129S-*Tac1*tm1.1(cre)Hze/J; Strain # 021877) breeders were obtained from The Jackson Laboratory (600 Main Street, Bar Harbor, ME USA 04609). Litters generated were separate and used randomly by the investigators. An adeno-associated virus (AAV$_5$-EF1a-DIO-ChETA-eYFP; qPCR titer of virus (vg/ml)=$3.9 \times 10^{12}$) that expresses ChR2-E123T Accelerated (ChETA) and the fluorescent protein eYFP was purchased from University of North Carolina (UNC vector core, Chapel Hill, NC, USA). The same serotype virus that expresses eYFP but lacks expression of the light-sensitive protein (AAV-EF1a-DIO-eYFP-WPRE-pA; UNC vector core, Chapel Hill, NC, USA) was also used at the same concentration for control experiments. Fentanyl citrate (50 µg/mL, Sandoz) was used with Health Canada exemption and obtained from the St. Michael's Hospital In-Patient Pharmacy.

## Surgical procedures

All surgeries were performed using standard aseptic techniques. Before surgery, mice were given pain medication: Anafen (5 mg/Kg), Dexamethazone (5 mg/Kg), and saline for a total volume of 1 ml

subcutaneously. Mice were anesthetized with isoflurane (2–3% in 100% oxygen) and placed in the prone position in a stereotaxic apparatus (model 940, KOPF instruments, Tujunga, CA, USA) with blunt ear bars where anesthesia was maintained through a nose cone. Adequate depth of anesthesia was determined via breathing and reflex responses to toe pinch and adjusted if necessary. Animal temperature was monitored through a rectal probe and was maintained at 36.5 °C via a heating pad (Kent Scientific corporation, Torrington, CT, USA). An incision was performed in the skin to expose the dorsal skull, which was then leveled horizontally between bregma and lambda. A small craniotomy was made to access the preBötzinger Complex. A needle (C315I-SPC Internal 33GA; P1 technologies,6591 Merriman Rd., Roanoke, VA, USA) containing the adeno-associated virus (AAV$_5$-EF1a-DIO-ChETA-eYFP) was slowly lowered in the preBötC 6.8 mm posterior, 1.2 mm lateral, and 6.4 mm ventral to bregma. Coordinates were chosen based on the Mouse Brain in Stereotaxic Coordinates (3$^{rd}$ Edition, Paxinos, and Franklin) combined with preliminary mapping of viral injections from our team. Needle was lowered with an infusion rate of 80 nl/min provided by a programmable syringe pump (Harvard Apparatus, Holliston, MA, USA) to avoid any clogging. Once in the targeted region, virus was infused at 50 nl/min reaching a total volume of 300 nl. Control animals were injected with a sham virus that lacks expression of the light-sensitive protein (AAV-EF1a-DIO-eYFP-WPRE-pA). Following infusion, the needle was maintained in position for 10 min before it was slowly removed and skin was sutured over the skull.

For unanesthetized/freely behaving experiments, an optic fiber was fixed over the preBötC following virus injection. Before the skin was sutured, two sterile stainless-steel screws (P1 Technologies, Roanoke, VA, USA) were implanted in the bone, one near the bregma and one near the preBötC craniotomy. A custom-made sterile cannula containing the optical fiber (200 μm, 0.5 NA; FP200ERT, Thor Labs, New Jersey, USA) was then positioned on top of the preBötC 6.8 mm posterior, 1.2 mm lateral, and 5.5 mm ventral to bregma. The cannula was then fixed in position resting on top of the skull using dental cement (Co-oral-ite dental MFG. CO., Diamond Spring, CA, USA). Cement was spread on both stainless-steel screws to firmly hold the cannula and skin was later sutured over it.

In the 3 days post-surgery, mice were given pain medication: Anafen (5 mg/Kg), Dexamethazone (5 mg/Kg), and saline for a total volume of 1 ml subcutaneously. Animals were kept for 2–4 weeks before experiments were performed to allow ChETA expression in the targeted region. Rostro-caudal histology of coronal medullary sections was performed to determine the expression of ChETA-eYFP in *Tac1* neurons. The location of the injection was confirmed by combining in situ hybridization for the expression of *Tacr1* (gene coding for neurokinin-1 receptors) and eYFP (marking ChETA). To determine whether ChETA expression was limited to Tac1-expressing cells, in situ hybridization was also performed for eYFP and *Tac1* genes.

## Optogenetic stimulation in anesthetized mice

Optogenetic stimulation was used to selectively activate tachykinin precursor 1 (*Tac1*) and vesicular-glutamate transporter 2 (*Slc17a6*) expressing cells in the preBötC. Three *Tac1*-IRES2-Cre-D and two *Slc17a6*-ires-cre ( threemales and 2 females) received the sham virus and served as controls while twenty *Tac1*-IRES2-Cre-D (14 males and 6 females) and five *Slc17a6*-ires-cre (3 males and 2 females) received the AAV$_5$-EF1a-DIO-ChETA-eYFP virus. Two or 4 weeks after surgery, mice were anesthetized with 2–2.5% isoflurane and were spontaneously breathing (50% oxygen gas mixture, balance nitrogen). Adequate depth of anesthesia was determined via breathing and reflex responses to toe pinch and adjusted if necessary. Animal temperature was monitored through a rectal probe and was maintained at 36.5 °C via a heating pad (Kent Scientific corporation, Torrington, CT, USA). Diaphragm muscle activity was recorded using bipolar electrodes sutured to the upper right abdominal wall adjacent to the diaphragm. Electromyography signals were amplified (AM Systems Model 1700, Sequim, Washington, USA), band-pass filtered (300–5000 Hz), integrated, and digitized at a sampling rate of 1000 Hz using PowerLab 4/26 acquisition system and LabChart Version 8 (ADInstruments, Colorado Spring, CO, USA).

Mice were placed in the prone position in a stereotaxic frame (model 940, KOPF instruments, Tujunga, CA, USA) with blunt ear bars where anesthesia was maintained through a nose cone. As this group did not have a fixed optical fiber over the preBötC, incision was again performed in the skin to expose the bregma and lambda. The dorsal skull was levelled horizontally between bregma and lambda and a small craniotomy was made to access the site of virus injection in the preBötC. An

optical fibre (200 µm, 0.5 NA; CFMC52L10, Thor Labs, New Jersey, USA), connected to a laser source (Laserglow technologies, North York, ON, Canada) was then positioned 6.8 mm posterior, 1.2 mm lateral, and 5.5 mm ventral to bregma. These coordinates positioned the optical fibre 0.2 mm dorsal to the preBötC. The location of the optical fibre was also confirmed by occasionally activating the laser (50 pulses of 20 ms repeated at 20 Hz, power 10 mW). Once breathing was stable (~10 min; average breathing rhythm = 20–30 breaths/min), laser stimulation (blue, wavelength = 473 nm) was performed using various settings. Settings included 20ms pulses repeated 10 times at frequencies of 5, 10, 20, 30, and 40 Hz and power of 1, 3, 5, and 10 mW. Duration of each stimulation train of 10 pulses (1.82, 0.92, 0.48, 0.32, 0.24 s) depended on the respective frequency (5, 10, 20, 30, and 40 Hz). For all stimulations, each train of 10 pulses was repeated five times with 1 s pause between them. A 3D Monte Carlo simulation platform was used to assess the theoretical spread of the laser light in brain tissue (*Liu et al., 2015*).

Following stimulation phases under control conditions, diaphragm activity was recorded for 10 min as a baseline sequence before mice were getting an intramuscular injection of the µ-opioid receptor agonist fentanyl (5 µg/Kg). 5 min were then given for fentanyl to elicit a respiratory rate depression and laser stimulation was once again performed using various settings. Settings included 20 ms pulses repeated 10 times with a frequency of 20, 30, and 40 Hz and power of 10 mW. For all stimulations, each train of 10 pulses had 1 s pause between them. Total duration of each photostimulation protocol was 1 min. Mice were then euthanized with the intracardiac injection of T-61 and the brain was harvested for post-mortem histology.

Data was extracted from LabChart Version 8 (ADInstruments, Colorado Spring, CO, USA) and exported to Microsoft Excel and GraphPad Prism 9 (Version 9.3.1; GraphPad Software) for analysis. For each laser stimulation, respiratory rate (breaths/min) and amplitude (volts) as well as inspiratory and expiratory durations were obtained from rectified diaphragm EMG recordings, averaged over the whole stimulation period, and normalized according to the preceding baseline period (rate and amplitude; an average of 1 min) or the average of the five preceding breathing cycles (inspiratory and expiratory durations). Since respiratory rate, amplitude and inspiratory/expiratory durations varied from one animal to another, these parameters were expressed as percentage changes in the values of the preceding baseline period. Figures with absolute values and means are provided in Supplementary Materials. Timing of the laser stimulation inside the respiratory cycle was also determined to measure the induced period. To do so, we calculated the duration between the initiation of inspiration and the start of laser stimulation and normalized it according to the preceding respiratory cycle.

## Optogenetic stimulation in freely behaving mice

Respiratory variables in freely behaving mice were recorded using the whole body, flow-through plethysmography (Buxco Electronics, DSI, New Brighton, Minnesota, USA). Two *Tac1*-IRES2-Cre-D and Two Slc17a6-ires-cre (3 males and 1 female) received the sham virus and served as controls while 10 *Tac1*-IRES2-Cre-D (7 males and 3 females) received the AAV$_5$-EF1a-DIO-ChETA-eYFP virus. Plethysmography chambers measured 21.5 cm in diameter and allowed enough room for mice to move freely during physiological recordings. Chambers were continuously ventilated using room air at a rate of 0.9 L/min (normoxia; F$_I$O$_2$=0.21) at room temperature. Mice were habituated to the plethysmography chamber and optic fiber cable for three days prior to experiments between 9:00 and 12:00 or 13:00 and 16:00. Experiments took place the next day during the same time of day as the habituation phase. On the day of the experiment, calibration of the system was performed by rapidly injecting 5 ml of air into the chamber with a syringe. Unanesthetized mice were gently handled to measure body temperature (Physitemp, Clifton, NJ, USA) and were tethered to the fiber optic cable and placed in the plethysmograph chamber. Animals were given 1 hr to acclimatize to the environment. Pressure changes inside the chamber were recorded with a pressure transducer, amplified (PS100W-2, EMKA Technologies, France), and digitized using PowerLab 4/26 in LabChart Version 8 (ADInstruments, Colorado Spring, CO, USA). Respiratory frequency ($f_R$) and tidal volume (V$_T$) were obtained from the plethysmograph signal. Barometric pressure, body temperature (Tb), chamber temperature, and humidity were measured to correct and standardize V$_T$ and values were expressed in ml BTPS (body temperature and pressure when saturated with water vapor) (*Drorbaugh and Fenn, 1955*; *Bartlett and Tenney, 1970*). Once the mouse appeared calm (respiratory rate ~150–200 breaths/min), laser stimulation (blue, wavelength = 473 nm) was performed using various settings under control

conditions. Settings included 20ms pulses repeated five times with a frequency of 30, 40, 50, 60, and 80 Hz and power of 10 mW to surpass the normal respiratory rate of the animal. Each train of five pulses was repeated for 10 s.

Following stimulation phases under control conditions, respiratory variables were recorded for 10 min as a baseline condition before mice were getting an intraperitoneal injection of either saline or the μ-opioid receptor agonist fentanyl (0.3 mg/Kg). This dose of fentanyl is considered a high dose in mice (*Fujii et al., 2019*). Five minutes were then given for fentanyl to create a respiratory rate depression and laser stimulation was once again performed using various settings. Settings included 20 ms pulses repeated five times with a frequency of 40 Hz and power of 10 mW. Each train of five pulses wase repeated for 10 s. Mice were then anesthetized with isoflurane (3% in 50% oxygen + 50% medical air), euthanized with the intracardiac injection of T-61 and the brain was harvested for post-mortem histology.

Data was extracted from LabChart Version 8 (ADInstruments, Colorado Spring, CO, USA) and exported to Microsoft Excel and GraphPad Prism 9 (Version 9.3.1; GraphPad Software) for analysis. For each laser stimulation, respiratory frequency, tidal volume, and inspiratory/expiratory durations were averaged over the whole stimulation period and normalized according to the preceding baseline period (frequency and tidal volume; average of 1 min) or the average of the five preceding breathing cycles (inspiratory and expiratory durations). These parameters were once again expressed as percentage changes infor the entire duration the values of the preceding baseline period.

## Respiratory and behavioral profiling

Whole-body plethysmography chambers with transparent platforms were mounted on a box with an HD 1080 P high-definition camera placed at the bottom of the box. Mouse movements were tracked from below for the entire duration of the experiments. Videos were recorded using Pinnacle Studio 24 MultiCam Capture software (Corel, Ottawa, Ontario, Canada), resized using Pinnacle Studio 24, and exported to EthoVision XT Version 14 (Noldus, Wageningen, the Netherlands) for analysis of movement parameters. Mouse velocity (cm/s) and activity (% of pixel change) were quantified and exported to Microsoft Excel for analysis in parallel with the respiratory parameters during stimulation in freely behaving mice. Video data was aligned with plethysmography recordings using video time stamps. The behavioral state of the animal was defined before each stimulation using respiratory recording and video monitoring. The animal was considered in a calm state if respiratory rate was ≤125% of baseline respiratory rate, velocity (cm traveled per second) was ≤0.1 cm/s, and when activity (% of pixel changed in frame) was ≤0.4%. The animal was considered in an active state if respiratory rate was >125% of baseline respiratory rate, velocity was >0.1 cm/s, and activity was >0.4%.

## In situ hybridization

To determine the expression of *Tac1*, *Slc17a6,* and *Oprm1* and confirm virus position through *eYFP* and *Tacr1* expression, in-situ hybridization was performed in C57BL/6 J, *Tac1*-IRES2-Cre-D, and Slc17a6-ires-cre mice which received a viral injection. Mice were perfused with phosphate-buffered saline (PBS) followed by formalin and the brain was harvested and placed into formalin solution overnight at room temperature. Brains were then soaked in 20% sucrose in PBS for 24 hr followed by 30% sucrose for 24 hr. Fixed brains were frozen using Tissue-Tek O.C.T. Compound (Sakura) and dry ice and stored at –80 °C. Coronal sections containing the preBötC were cut at 20 μm thickness using a cryostat (Model CM3050S, Leica Biosystems, Wetzlar, Germany) and mounted on superfrost plus slides (VWR International, Radnor, PA, USA). Sections were scanned using the Axio Scan.Z1 slide scanner (ZEISS, Germany) to confirm optical fibre location. The manufacturers' protocol was used to perform in-situ hybridization (RNAscope Multiplex Fluorescent Reagent v2 Assay, Advanced Cell Diagnostics, Newark, California, USA) and sections were counterstained with DAPI. Target probes used included combinations of Mm-*Tac1* (Cat No. 410351-C2) targeting *Tac1* gene mRNA, Mm-*Slc17a6* (Cat No. 319171-C2) targeting *Slc17a6* gene mRNA, Mm-*Oprm1* (Cat No. 315841) targeting *Oprm1* gene mRNA, Mm-*Tacr1* (Cat No. 428781) targeting NK1 receptor gene mRNA and Mm-*eYFP* (Cat No. 551621-C3) targeting the gene coding for eYFP. Genes are indicated in italics and with the first letter in the capital in this manuscript. Tissue sections 40 μm apart were scanned using the Axio Scan.Z1

slide scanner (ZEISS, Germany). As previously described, to identify sections containing the preBötC, the Mouse Brain in Stereotaxic Coordinates (3rd Edition, Paxinos, and Franklin) was consulted, and anatomical markers used included (1) the nucleus tractus solitarius, (2) the nucleus ambiguus, (3) the facial nucleus, (4) the hypoglossal nucleus, (5) the external cuneate nucleus as a surface landmark, and (6) the overall shape of the section. To quantify mRNA expression, 3 brains were used and 3 sections from each brain containing the preBötC (extending approximately 240 µm rostral-caudal) were exported from Zen (ZEISS, Germany) to Adobe Illustrator (Creative Suite 5, Adobe) where regions of interest were drawn. Images were then exported to Fiji (ImageJ) for counting. Cells count from the three sections of each brain were averaged. Counts were obtained for total DAPI, *Tac1* mRNA, *Slc17a6* mRNA, and *Oprm1* mRNA. mRNA expression was expressed either as a percentage of total DAPI cells, total *Tac1* cells, total *Slc17a6* cells, or total *Oprm1* cells. Images were produced using ZEN (ZEISS, Germany).

## Statistics

Statistical analysis and graphs were performed using GraphPad Prism 9 (Version 9.3.1; GraphPad Software). Figures were prepared using Adobe Illustrator (Creative Suite 5, Adobe). Data in all figures and text are reported as means ± standard error of the mean (SEM) with individual data points also displayed. Data from male and female pups were combined as preliminary statistical analyses revealed no sex specific responses to laser stimulation. Data was first tested for normality by using a Shapiro-Wilk test. Normally distributed data were analyzed with ANOVAs (one-way ANOVA, two-way ANOVA, or two-way repeated measures ANOVA) or a mixed effects analysis if there were missing values. Factors considered in the analyses were the effect of group (control, *Tac1*, *Slc17a6*), laser stimulation, laser power (mW), and stimulation phase. A simple linear regression was used to determine the relationships between percentage changes in respiratory rate, velocities, or baseline respiratory rates. When ANOVA results indicated that a factor (or a factorial interaction) was significant ($p \leq 0.05$), a Tukey's multiple comparisons test or Šidàk's multiple comparisons test was performed for *post hoc* analysis. ANOVA results are mainly reported in the text while results from *post hoc* tests are reported in figures using symbols.

## Additional information

### Funding

| Funder | Grant reference number | Author |
| --- | --- | --- |
| CIHR | | Jean-Philippe Rousseau |
| CIHR Project Grant | | Gaspard Montandon |

The funders had no role in study design, data collection and interpretation, or the decision to submit the work for publication.

### Author contributions

Jean-Philippe Rousseau, Conceptualization, Data curation, Software, Formal analysis, Validation, Investigation, Visualization, Methodology, Writing - original draft, Writing - review and editing; Andreea Furdui, Formal analysis, Methodology; Carolina da Silveira Scarpellini, Data curation, Software, Formal analysis, Visualization, Methodology; Richard L Horner, Conceptualization, Supervision, Funding acquisition, Investigation, Writing - original draft, Writing - review and editing; Gaspard Montandon, Conceptualization, Resources, Software, Formal analysis, Supervision, Funding acquisition, Validation, Investigation, Visualization, Methodology, Writing - original draft, Project administration, Writing - review and editing

### Author ORCIDs

Carolina da Silveira Scarpellini (iD) https://orcid.org/0000-0001-5576-3468
Gaspard Montandon (iD) http://orcid.org/0000-0003-3587-4472

### Ethics

All procedures were carried out in accordance with the recommendations of the Canadian Council on Animal Care and were approved by St. Michael's Hospital animal care committee (animal use protocols #981 and #988).

### Decision letter and Author response

Decision letter https://doi.org/10.7554/eLife.85575.sa1
Author response https://doi.org/10.7554/eLife.85575.sa2

---

## Additional files

### Supplementary files
• MDAR checklist

### Data availability

All data generated are included in the manuscript and supporting files. Source data files are provided for all Figures.

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
