## [Editor Report]

This important study expands our understanding of the neuronal composition and functions of the brainstem region called the preBötzinger complex (preBötC)- the site where the rhythm of breathing in mammals originates. The authors identify from gene expression mapping a small population of primarily excitatory neurons expressing the gene (Tac 1) for Substance P and also, in many cells, co-expressing the mu-opioid receptor gene (Oprm1) within this region. From targeted optogenetic photostimulation studies in anesthetized and freely-behaving mice in vivo, the authors present solid evidence that the Tac1-expressing neurons play an important role in modulating the inspiratory rhythm, are capable of overcoming respiratory depression by opioid drugs (fentanyl in these studies), and also can induce locomotion.

---

## [Decision Letter]

**Decision letter after peer review:**

Thank you for submitting your article "A tachykinin precursor 1 medullary circuit promoting rhythmic breathing" for consideration by *eLife*. Your article has been reviewed by 3 peer reviewers, including Jeffrey C Smith as Reviewing Editor and Reviewer #1, and the evaluation has been overseen by Kate Wassum as the Senior Editor. The following individual involved in review of your submission has agreed to reveal their identity: Luis Rodrigo Hernandez-Miranda (Reviewer #2).

Essential revisions:

1) Evidence for some of the most important conclusions, particularly those regarding locomotion and state-dependence that suggest Tac1 neurons represent a "new" functional population of preBötC neurons, is incomplete/inadequate. The experimental evidence presented suggesting a novel respiratory role of these neurons, a state-dependence to their function, or a direct role in opioid-induced respiratory depression is limited. Furthermore, the evidence for preBötC neuron site-specificity for promoting locomotion is also limited. The addition of new experiments and analyses that better support these suggestions and the authors' conclusions is needed.

2) The authors should provide more detailed anatomical reconstructions of the spatial extent including rostrocaudal distribution of the virally transduced expression of ChETA-eYFP in Tac1 neurons to more clearly convey the regions in which Tac1 neurons were expressing ChETA-eYFP and potentially photostimulated. Also, regarding anatomical reconstruction, where cell counts are provided for various labeling patterns (Figures 3 and 7), please indicate the number of serial sections used in each case. How have the authors determined that their representations of cell count accurately convey the actual labeling patterns through the total region of interest?

3) The claim of state dependence is not adequately justified, because the distinction between "calm" and "active" states has little rationale and seems arbitrarily based only on respiratory frequency. Since the only criterion is respiratory rate, if the data are separated in this way, the terminology used should reflect this. It would be more appropriate to remove "calm or active" throughout the manuscript and more appropriately state that the respiratory effects were frequency dependent, based upon the respiratory frequency at the time of stimulation, etc. Moreover, this analysis seems misleading since it is not surprising that the stimulations appear to have a larger effect on breathing when baseline frequency was slow (calm state) when the effect on breathing is quantified as a % change from baseline. To further illustrate this, since mice have a maximum frequency they can breathe, there is likely a ceiling effect, e.g., if the mouse is already breathing close to that maximum at baseline (active state), it would not be surprising that stimulation would be unable to increase it further.

4) The authors' conclusion that photostimulating specifically preBötC Tac1-expressing neurons induces locomotion needs to be strengthened by incorporating additional control studies on the site specificity of this effect since Tac1-expressing neurons and the viral-induced expression of ChETA in these neurons is not strictly confined to the preBötC region, and nearby regions of the medulla oblongata that are labeled are also locomotion producing. The authors suggest in the Discussion that non-site-specific activation of Tac1 neurons is unlikely to account for the induction of locomotor behavior, but how has this been established as a control? The optical fiber used is large (200 µm) relative to the regions of interest as well as the high laser power (10 mW), probably illuminating large areas of the medullary ventrolateral and ventromedial reticular formation. Does repositioning the optical fiber fundamentally change the results? Related to this, the authors state (p. 7, line 271) that position was confirmed through continuous high-frequency laser stimulation at 10 mW. Optical fiber position would ideally be confirmed by post hoc anatomical reconstruction. Was this routinely done?

5) The authors should discuss the possibility of off-target labeling of neurons known to be involved in the modulatory control of breathing and potentially expressing Tac1. The large viral injection volume (300 nl) seems much too large to achieve specific targeting of the preBötC. This off-target labeling could include C1 neurons of the RVLM and IVLM that are glutamatergic and some of which express substance P (see also the comments of Reviewer #2 regarding other possible off-target neuronal populations). The authors should discuss this issue and describe how site specificity was achieved in the experiments. Please also provide the viral titers used.

6) The authors need to address all other issues listed by the reviewers in their recommendations for the authors.

7) If you have not already done so, please include a key resource table and clarify the distribution of sexes across groups.

*Reviewer #1 (Recommendations for the authors):*

This valuable study expands the current understanding of the neuronal composition and functions of the region of the brainstem called the preBötzinger Complex (preBötC)-the site where the rhythm of breathing in mammals originates. The authors identify from gene expression mapping a small population of mostly excitatory neurons expressing the gene (Tac1) for Substance P, and also, in many cells, co-expressing the mu-opioid receptor gene (Oprm1) within this region. From targeted optogenetic photostimulation studies in anesthetized and freely-behaving mice in vivo, the authors present solid evidence that the Tac1-expressing neurons play an important role in modulating the inspiratory rhythm, as well as are capable of overcoming respiratory depression by opioid drugs (fentanyl in these studies), and also triggering locomotion, although this latter claim would be strengthened by incorporating additional control studies on the site specificity of this effect.

Advantages of the study include:

The authors use a nice combination of gene expression mapping by mRNA in situ hybridization, transgenic mouse optogenetics employing virally transduced channelrhodopsin variant (ChETA- eYFP) for selective photostimulation of Tac1-expressing neurons, electrophysiological, and behavioral approaches.

Solid evidence is presented demonstrating for the first time that photostimulation of Tac1 neurons can trigger inspiration and modulate the respiratory rhythm when stimulated in live and freely behaving rodents. Also, their results from behavioral measurements demonstrate that stimulation of the Tac1 preBötC cells promotes substantially respiratory rate during the "calm" state, but only moderately during the "active" state, defined respectively, by the respiratory frequency at rest and during activity in freely behaving mice.

Evidence is presented that photostimulation of Tac1-expressing neurons promotes locomotor behaviors. Thus, the authors suggest that Tac1-expressing cells in respiratory circuits may link breathing to motor behaviors, which may play an important role in behavioral responses, such as the fight-or-flight response, associated with locomotor activity and cardio-respiratory responses.

Valuable results are presented showing that the subpopulation of preBötC glutamatergic neurons expressing the Tac1 gene that promotes breathing also expresses the mu-opioid receptor gene Oprm1. Furthermore, the Tac1 cells when photostimulated reverse respiratory depression by the opioid drug fentanyl, with potential therapeutic implications.

Disadvantages of the study include:

The authors do not provide sufficiently detailed anatomical reconstructions of the spatial extent of the virally transduced expression of ChETA-eYFP in Tac1 neurons to more clearly convey the regions in which Tac1 neurons were labeled and potentially photostimulated. How the authors established site-specificity for the virally transduced ChETA-eYFP in Tac1 neurons is not adequately described.

The authors' suggestion that photostimulating specifically preBötC Tac1-expressing neurons induces locomotion needs to be strengthened by incorporating additional control studies on the site-specificity of this effect since Tac1-expressing neurons and the viral-induced expression of ChETA in these neurons is not strictly confined to the preBötC region and includes nearby regions of the medulla oblongata that are expressing ChETA-eYFP are also locomotion producing areas.

1) The authors should provide more detailed anatomical reconstructions of the spatial extent including rostrocaudal distribution of the virally transduced expression of ChETA-eYFP in Tac1 neurons to more clearly convey the regions in which Tac1 neurons were expressing ChETA-eYFP and potentially photostimulated. How the authors established site-specificity for the virally transduced ChETA-eYFP in Tac1 neurons is not adequately described.

2) Related to the above anatomical reconstruction issue, where cell counts are provided for various labeling patterns (Figures 3 and 7), please indicate the number of serial sections used in each case. How have the authors determined that their representations of cell count accurately convey the actual labeling patterns throughout the total region of interest?

3) The authors' conclusion that photostimulating specifically preBötC Tac1-expressing neurons induces locomotion needs to be strengthened by incorporating additional control studies on the site specificity of this effect since Tac1-expressing neurons and the viral-induced expression of ChETA in these neurons is not strictly confined to the preBötC region, and nearby regions of the medulla oblongata that are labeled are also locomotion producing. The authors suggest in the Discussion that non-site-specific activation of Tac1 neurons is unlikely to account for the induction of locomotor behavior, but how has this been established as a control? The optical fiber used is large (200 µm relative to the regions of interest) as well as the laser power (10 mW), probably illuminating large areas of the medullary ventrolateral reticular formation. Does repositioning the optical fiber fundamentally change the results? Related to this, the authors state (p. 7, line 271) that position was confirmed through continuous high-frequency laser stimulation at 10 mW. Optical fiber position would ideally be confirmed by post hoc anatomical reconstruction. Was this routinely done?

*Reviewer #2 (Recommendations for the authors):*

Breathing (or respiration) is a complex motor behavior produced by dedicated brainstem neurons. One group of such neurons is the preBötzinger complex that contains cells responsible for generating the respiratory rhythm. Neurons in the preBötzinger complex are heterogeneous and the physiology of such cells are not yet fully understood. In this work, the authors set to characterize a subgroup of preBötzinger neurons in respiration using a variety of anatomical, physiological, and behavioral assays. The reported results show that the investigated neurons are not only key for respiratory rhythmic patterns but also critical for overcoming the characteristic respiratory depression caused by opioid drugs. This is perhaps the most exciting part of this work. Furthermore, the authors provide evidence that might also link these neurons to locomotion.

Strengths/weaknesses

In my view, this work is well-conducted and most of the conclusions are supported by strong experimental data. One methodological point that deserves attention is related to potential off-target effects of the performed viral injections. The injected area contains in addition to the studied cells, other populations of neurons known to contribute to respiration. The authors correctly discuss potential off-targets for locomotion but perhaps those in connection to respiration are less discussed.

The work by Rousseau and colleagues entitled "A tachykinin precursor 1 medullary circuit promoting rhythmic breathing" centers at the characterization of a subpopulation of preBötzinger neurons expressing substance P (Tac1). These neurons are key in the generation of the respiratory rhythm and might constitute a link between respiration and locomotion. This work is thus valuable for the respiratory research field.

In general, I have no major problems with this manuscript. However, there is an issue that I believe needs small anatomical clarification or at least more discussion.

Can the authors exclude the transduction of PiCO neurons (dorsally located to the nucleus ambiguous) and A1 adrenergic neurons by their viral infections? From Figures2e and 4h, it seems that several neurons in the "PiCO" and "A1" areas are also transduced. A1 (TH^+^) and PiCO (ChAT+) neurons share (or have a history of) vGlut2 expression and might be also Tac1+. It might be worth discussing potential off-target transductions in these areas and their potential influence in the described physiological results. This is particularly relevant as some changes are seen in the early expiration (or postinspiration) phase.

*Reviewer #3 (Recommendations for the authors):*

The preBotzinger Complex in the ventral medulla is critical for generating respiratory rhythm. Rhythmic activity of this bilaterally distributed network of excitatory and inhibitory interneurons is regulated by many neuromodulators including Substance P, a neuropeptide encoded by the Tac1 gene. Substance P is known to facilitate the respiratory rhythm and counteract the depressive effects of opioids in vitro. However, how Substance P regulates network function in vivo is less well understood. Using an optogenetic approach to allow activation of Tac1-expressing neurons with light, Rousseau et al., investigate how manipulating the activity in these neurons within or near the pre-Bot affects breathing and locomotion. The authors show that Tac1 neurons in the pre-BötC are also glutamatergic, and that stimulation of this Tac1+ glutamatergic subpopulation in anesthetized mice elicits similar, but weaker, effects on breathing compared to stimulation of glutamatergic pre-Bot neurons in general. A limitation is that the experiments presented here do not determine the relative roles of substance P and glutamatergic signaling, and therefore the functional significance of Tac1 expression in this subgroup of pre-Bot neurons remains unclear. Stimulation of Tac1 pre-BotC neurons in awake mice also increases average breathing frequency over the course of the stimulation period, but breathing patterns are highly variable during stimulation and effects on breathing are associated with changes in locomotor activity, which could indicate that some changes in breathing are secondary to stimulation induced changes in behavior. The authors suggest a potential state-dependent role of Tac1 pre-Bot neurons because average changes in breathing frequency relative to baseline (% change) are more pronounced when stimulations occurred during periods of slow vs fast baseline breathing (referred to as "calm" vs. "active" states). However, this interpretation is limited by unclear justification for parameters used to define these states and by misleading data analysis. The study also demonstrates that the ability of Tac1 pre-Bot neurons to increase breathing rate remains present in the context of opioid-induced respiratory depression. A strength of the study is ISH experiments demonstrating that many Tac1 pre-Bot neurons express Oprm1. However, the role of these neurons in opioid induced respiratory depression and the significance of this role relative to other pre-Bot neurons remains inferred or unknown since experiments to e.g. knockout Oprm1 from Tac1 neurons are not performed.

This manuscript investigates on the role of Tac1 expressing neurons within the preBötzinger complex of the ventral brainstem. The authors show that some of the glutamatergic neurons in the pre-BötC co-express Tac1, the precursor to substance P. Compared to the known effects of glutamatergic pre-Bot neurons in general, stimulation of this Tac+ glutamatergic subpopulation elicits similar, but weaker, effects on breathing in anesthetized mice. Stimulation of Tac1 pre-BotC neurons in awake mice also increases average breathing frequency over the course of a ~10s stimulation period, with breathing patterns being highly variable and associated with changes in locomotor activity. Increases in average breathing frequency relative to baseline (% change) are more pronounced when stimulations occurred during periods of slow vs fast baseline breathing (referred to as "calm" vs. "active" states), which is interpreted as state-dependence. The study also demonstrates that the ability of Tac1 pre-Bot neurons to increase breathing rate remains present in the context of opioid-induced respiratory depression. The observations that some pre-BotC neurons express Tac1, that this subpopulation contributes to respiratory function, and that many of these Tac1 neurons express Oprm1 are important. However, evidence indicating a unique respiratory role of these neurons, a state-dependence to their function, or a direct role in opioid induced respiratory depression is limited, reducing the overall significance of the study.

Specific Comments:

Line 50: Is this a new population? If so, how? Are these Tac1 neurons functionally different than other glutamatergic pre-BotC neurons?

Line 67: This seems overstated – What is evidence that blocking SP alters rhythmic breathing? The role of Tac1 or NK1R expressing neurons is much different that the role of SP signaling. This should be considered throughout the manuscript.

Line 129: An injection volume of 300nl seems much too large to achieve specific targeting of the pre-BotC. The authors should comment on this and describe how specificity was achieved in their experiments. Also, what was infused at 80nl/min during insertion of the needle? What type of needle was used?

Line 297: What is baseline respiratory rate in anesthesia? What is anesthesia used? It would be helpful to mention this in Results. Is there a difference in viral expression at 2 vs 4 wks that explains the different results at these two timepoints?

Line 304: Breathing was significantly affected when normalized to baseline, but not when raw breathing frequency was compared. This should be noted in the results.

Line 346: Figure 2e,f – The images seem to show clear examples of cells that express EYFP that DO NOT express Tac1… If so, this either suggests that the ISH for Tac1 is not labelling all Tac1 neurons, or the viral vector is not 100% specific for Tac1 neurons. Please comment.

Line 430: suggest alternative to "in vivo". Studies in anesthetized mice also considered in vivo.

Line 431: RE the distinction between calm and active "states": "Based on baseline respiratory rate" – please elaborate on this here. How can baseline breathing rate be more or less than 125% baseline breathing rate? This seems somewhat arbitrary and needs justification. Since the only criteria is respiratory rate, if the data are separated in this way, the terminology used should reflect this. It would be more appropriate to remove "calm or active" throughout the manuscript and more appropriately state that the respiratory effects were frequency dependent, based upon the respiratory frequency at the time of stimulation, etc. Moreover, this analysis seems misleading since it is not surprising that the stimulations appear to have a larger effect on breathing when baseline frequency was slow (calm state) when the effect on breathing is quantified as a % change from baseline. To further illustrate this – since mice have a maximum frequency they can breathe, there is likely a ceiling effect. E.g. if the mouse is already breathing close to that maximum at baseline (active state), it would not be surprising that stimulation would be unable to increase it further.

Line 437: The raw frequency values are not included in the supplemental figures as stated in the text. This should be included.

Line 442: Again, this may not be surprising if mice have a maximum frequency they can breathe. E.g. if they are already breathing close to that maximum at baseline, stimulation will be unable to increase it further.

Why were stimulations delivered in 200 ms "bouts"?

Figure 5- what about other breath parameters – e.g. amplitude/volume that seem to change with stimulations in awake mice in addition to frequency?

Figure 6d,e, – velocity and activity should not be expressed as a % baseline since in both cases the baseline value is ~0. Raw values would be more appropriate i.e. cm/s.

Line 483-486: cannot make this conclusion based on the data presented, because control mice did not move fast or breath fast during the recordings. Therefore, the data do not demonstrate whether or not fast breathing and movement are also correlated in control mice.

Line 496: Should 38% be considered a "small fraction"?

Discussion Line 588: "Tac1 role on breathing depends on the state of the animal (calm vs. active), and therefore the overall excitability of the respiratory network, as a minimal effect of Tac1 stimulation was observed when the animal was behaviorally active." The claim of state-dependence does not seem justified. The distinction between calm and active states has little rationale and is based only on respiratory frequency.

Line 696: Entrainment of breathing by Tac1 neurons is not demonstrated here.

[Editors’ note: further revisions were suggested prior to acceptance, as described below.]

Thank you for resubmitting your work entitled "A tachykinin precursor 1 medullary circuit promoting rhythmic breathing" for further consideration by *eLife*. Your revised article has been evaluated by Kate Wassum (Senior Editor) and a Reviewing Editor.

The manuscript has been improved, but there are some remaining issues that need to be addressed, as outlined below:

Please revise the text according to each of the comments by Reviewer #3, which are essential to improve the interpretation and readability of this revised version of the manuscript. This reviewer's suggestion for revision of the manuscript Tile is appropriate. Please proofread the Abstract for grammatical errors and content (e.g., the last sentence).

*Reviewer #1 (Recommendations for the authors):*

The authors' revisions, including additional experimental data addressing the main concerns raised in the original review, have supported their main conclusions regarding the role of preBötC excitatory Tac1-expressing neurons and have substantially improved the manuscript. The inclusion of more detailed anatomical reconstructions of the spatial extent, including rostrocaudal distribution of the virally transduced expression of ChETA-eYFP in Tac1 neurons and the new simulations of laser light spatial power density profiles, provides essential information on the regions in which Tac1 neurons were expressing ChETA-eYFP and potentially photostimulated under the experimental conditions. The additional caveats about the site-specificity of the photostimulation behavioral responses included in the Discussion are appropriate and improve clarity.

*Reviewer #3 (Recommendations for the authors):*

The authors have made important revisions to the manuscript and have included additional data that has improved the study. But, the paper would still benefit from additional revisions.

Title – In this reviewer's opinion the data presented do not describe a "circuit" but characterize the function of a group of neurons in a single location. This may be better reflected by a title such as "Medullary tachykinin precursor 1 neurons contribute to rhythmic breathing".

Line 49-50: It seems somewhat misleading to state that a "new population" has been identified. As the authors state elsewhere, these neurons are very likely the same glutamatergic neurons that have been previously manipulated and functionally described using Vglut2-cre, Dbx1-cre, etc. The new insight is that some of these neurons express Tac1 and manipulating them has very similar consequences for breathing as manipulating the larger glutamatergic preBotC population.

Line 73: This is still overstated. There is very little data that substance P is "pivotal" for respiratory rhythm, and this could be confusing for the reader as it conflates the role of the neurons with the role of the neuropeptide. NK1-expressing neurons are pivotal, but this does not indicate a critical role for substance P.

Line 151-152: "Tac1 cells were only stimulated after virus incubated for 4 weeks, but not 2 weeks (n=25)" Do you mean it was only effective at 4 weeks?

Line 442-458: This section is somewhat confusing and again conflates the function of substance p/NK1-r signaling with the function of glutamatergic preBotC neurons that express Tac1.

In general, the writing could be further improved. There are numerous typos, inconsistent use of tense, sentences with missing words, repetitions, etc. For example, the impact statement includes "neurons in the medulla" twice within the same sentence.

It appears the reference list needs updating since some citations in the text do not appear in the bibliography.

---

## [Author Response]

Essential revisions:1) Evidence for some of the most important conclusions, particularly those regarding locomotion and state-dependence that suggest Tac1 neurons represent a "new" functional population of preBötC neurons, is incomplete/inadequate. The experimental evidence presented suggesting a novel respiratory role of these neurons, a state-dependence to their function, or a direct role in opioid-induced respiratory depression is limited. Furthermore, the evidence for preBötC neuron site-specificity for promoting locomotion is also limited. The addition of new experiments and analyses that better support these suggestions and the authors' conclusions is needed.

We have addressed the issues regarding state dependency and site specificity of ChETA-eYFP expression linked to breathing control and locomotor. Please see responses below.

2) The authors should provide more detailed anatomical reconstructions of the spatial extent including rostrocaudal distribution of the virally transduced expression of ChETA-eYFP in Tac1 neurons to more clearly convey the regions in which Tac1 neurons were expressing ChETA-eYFP and potentially photostimulated. Also, regarding anatomical reconstruction, where cell counts are provided for various labeling patterns (Figures 3 and 7), please indicate the number of serial sections used in each case. How have the authors determined that their representations of cell count accurately convey the actual labeling patterns through the total region of interest?

ChETA expression throughout the medulla and laser diffusion. Optogenetics allow stimulation of discrete populations of neurons. By combining virus injection in the preBötC, local expression of ChETA and focused laser photostimulation, we can ensure that only preBötC *Tac1* neurons are activated. To this aim, we have included in-situ hybridization sections of the rostro-caudal distribution of ChETA-eYFP and *Tac1* neurons in the medulla (Supplementary Figure 1). These sections contain various respiratory circuits around the location of the optic fiber. Using a 3D Monte Carlo simulation platform, we determined that a laser power of 5mW (the minimal power to produce an effect on breathing) produced a power of 5mW/mm^2^ in a region 470 µm beyond the optic fiber (See Supplementary Figure 1). This power density of 5mW/mm^2^ is the light power density required to achieve half-maximal ChETA activation (EPD50 or effective power density for 50% activation). According to our histology, ChETA is almost exclusively found within a small region including the preBötC and the rostro-ventral lateral medulla.

For the effect of *Tac1* preBötC stimulation on locomotion, we used a power 10 mW which increases the power density of the laser and enlarges the region where the laser light can activate ChETA (Supplementary Figure 4). In-situ hybridization showed that *eYFP*-ChETA and *Tac1* are highly expressed in the preBötC, moderately expressed in the CVLM, the LPGi, and the BotC, which are regions not known to be involved in locomotion. Considering the low expression in regions away from the preBötC, and the diffusing and fading light beyond the preBötC, we think that photostimulation mostly acts on preBötC neurons. However it cannot be excluded than other regions are stimulated and it is now addressed in the Discussion.

Cell counts for Tac1, Vglut2, and Oprm1. Regarding the cell count for the labeling patterns (Figures 3 and 7), cell count was done on 3 sections of each brain for a total of 3 mice. We then averaged the numbers from the 3 sections for each animal and inserted in the figure. This is further explained in the revised text and details can be found in the excel file for data of figures 3 and 7. Regions of interest were drawn based on the Mouse Brain in Stereotaxic Coordinates (3^rd^ Edition, Paxinos and Franklin). Cell counts were similar in most sections of the same animal and it accurately represented the expression of *Tac1* cells in the preBötC.

3) The claim of state dependence is not adequately justified, because the distinction between "calm" and "active" states has little rationale and seems arbitrarily based only on respiratory frequency. Since the only criterion is respiratory rate, if the data are separated in this way, the terminology used should reflect this. It would be more appropriate to remove "calm or active" throughout the manuscript and more appropriately state that the respiratory effects were frequency dependent, based upon the respiratory frequency at the time of stimulation, etc. Moreover, this analysis seems misleading since it is not surprising that the stimulations appear to have a larger effect on breathing when baseline frequency was slow (calm state) when the effect on breathing is quantified as a % change from baseline. To further illustrate this, since mice have a maximum frequency they can breathe, there is likely a ceiling effect, e.g., if the mouse is already breathing close to that maximum at baseline (active state), it would not be surprising that stimulation would be unable to increase it further.

This is a valid point. To better define active and calm states, we classified states according to velocity and activity using video tracking. An active state was defined when velocity was >0.1 cm/s and when activity (% of pixel changed in frame) was < 0.4%. When velocity and activity were lower than these values, the animal was considered in calm state. It is now described in Methods and in Figure 6.

4) The authors' conclusion that photostimulating specifically preBötC Tac1-expressing neurons induces locomotion needs to be strengthened by incorporating additional control studies on the site specificity of this effect since Tac1-expressing neurons and the viral-induced expression of ChETA in these neurons is not strictly confined to the preBötC region, and nearby regions of the medulla oblongata that are labeled are also locomotion producing. The authors suggest in the Discussion that non-site-specific activation of Tac1 neurons is unlikely to account for the induction of locomotor behavior, but how has this been established as a control? The optical fiber used is large (200 µm) relative to the regions of interest as well as the high laser power (10 mW), probably illuminating large areas of the medullary ventrolateral and ventromedial reticular formation. Does repositioning the optical fiber fundamentally change the results? Related to this, the authors state (p. 7, line 271) that position was confirmed through continuous high-frequency laser stimulation at 10 mW. Optical fiber position would ideally be confirmed by post hoc anatomical reconstruction. Was this routinely done?

We provided a detailed anatomical reconstruction of locomotor nuclei expressing ChETA-eYFP in Supplementary Figure 4. The 3D Monte Carlo simulation shows that photostimulation of 10mW spreads beyond the preBötC rostrocaudally and could affect some motor nuclei delineated. However, ChETA-eYFP expression remains limited the region of the preBotC. This is addressed in the revised text.

We performed histology for each experiment to determine the position of the optic fiber marked with fluorescent staining. An example is shown in Figure 2. The combination of ChETA expression and optic fiber position ensured that mostly preBötC cells were stimulated. However, we cannot exclude that other regions of the medulla were stimulated.

5) The authors should discuss the possibility of off-target labeling of neurons known to be involved in the modulatory control of breathing and potentially expressing Tac1. The large viral injection volume (300 nl) seems much too large to achieve specific targeting of the preBötC. This off-target labeling could include C1 neurons of the RVLM and IVLM that are glutamatergic and some of which express substance P (see also the comments of Reviewer #2 regarding other possible off-target neuronal populations). The authors should discuss this issue and describe how site specificity was achieved in the experiments. Please also provide the viral titers used.

We provide a detailed anatomical reconstruction of nuclei involved in control of breathing and potentially expressing ChETA-eYFP in Supplementary Figure 1. According to the 3D Monte Carlo simulation platform, RVLM and IVLM containing C1 neurons could be affected by the laser power of 5 mW. We are now addressing this possibility in the Discussion.

qPCR titer of virus (vg/ml) = 3.9 x 10^12^. This is now mentioned in the revised text.

6) The authors need to address all other issues listed by the reviewers in their recommendations for the authors.

The authors have addressed all the concerns raised in their reports.

7) If you have not already done so, please include a key resource table and clarify the distribution of sexes across groups.

Distribution of male and females across all groups has been clarified in the text.

Reviewer #1 (Recommendations for the authors):1) The authors should provide more detailed anatomical reconstructions of the spatial extent including rostrocaudal distribution of the virally transduced expression of ChETA-eYFP in Tac1 neurons to more clearly convey the regions in which Tac1 neurons were expressing ChETA-eYFP and potentially photostimulated. How the authors established site-specificity for the virally transduced ChETA-eYFP in Tac1 neurons is not adequately described.

See response above.

2) Related to the above anatomical reconstruction issue, where cell counts are provided for various labeling patterns (Figures 3 and 7), please indicate the number of serial sections used in each case. How have the authors determined that their representations of cell count accurately convey the actual labeling patterns throughout the total region of interest?

Regarding the cell count for the labeling patterns (Figures 3 and 7), cell count was done on 3 sections for each animal for 3 mice in total. The cell numbers from the 3 sections were average for each animal and used in bar graphs. This is further explained in the revised text and details can be found in the excel file for data of Figures 3 and 7. Regions of interest were drawn based on the Mouse Brain in Stereotaxic Coordinates (3^rd^ Edition, Paxinos and Franklin). x.

3) The authors' conclusion that photostimulating specifically preBötC Tac1-expressing neurons induces locomotion needs to be strengthened by incorporating additional control studies on the site specificity of this effect since Tac1-expressing neurons and the viral-induced expression of ChETA in these neurons is not strictly confined to the preBötC region, and nearby regions of the medulla oblongata that are labeled are also locomotion producing. The authors suggest in the Discussion that non-site-specific activation of Tac1 neurons is unlikely to account for the induction of locomotor behavior, but how has this been established as a control? The optical fiber used is large (200 µm relative to the regions of interest) as well as the laser power (10 mW), probably illuminating large areas of the medullary ventrolateral reticular formation. Does repositioning the optical fiber fundamentally change the results? Related to this, the authors state (p. 7, line 271) that position was confirmed through continuous high-frequency laser stimulation at 10 mW. Optical fiber position would ideally be confirmed by post hoc anatomical reconstruction. Was this routinely done?

See above.

Reviewer #2 (Recommendations for the authors):Can the authors exclude the transduction of PiCO neurons (dorsally located to the nucleus ambiguous) and A1 adrenergic neurons by their viral infections? From Figures2e and 4h, it seems that several neurons in the "PiCO" and "A1" areas are also transduced. A1 (TH^+^) and PiCO (ChAT+) neurons share (or have a history of) vGlut2 expression and might be also Tac1+. It might be worth discussing potential off-target transductions in these areas and their potential influence in the described physiological results. This is particularly relevant as some changes are seen in the early expiration (or postinspiration) phase.

We provide a detailed anatomical reconstruction of nuclei involved in control of breathing (including the PiCo and A1 region) and potentially expressing ChETA-eYFP in Supplementary figure 1. According to the 3D Monte Carlo simulation platform, PiCo could be affected by the laser power of 10mW, but not 5mW, at which it produces inspiration. Importantly, ChETA-eYFP expression was absent in this region. Off target transduction and their effect on breathing is addressed in the revised text.

Reviewer #3 (Recommendations for the authors):Specific Comments:Line 50: Is this a new population? If so, how? Are these Tac1 neurons functionally different than other glutamatergic pre-BotC neurons?

Tac1 expressing cells have been defined in other nuclei and are related to behaviors like breathing, pain or locomotion, but this is the first study addressing the function of Tac1 preBötC neurons in the control of breathing. How the function of these neurons differs from other glutamatergic cells is unknown.

Line 67: This seems overstated – What is evidence that blocking SP alters rhythmic breathing? The role of Tac1 or NK1R expressing neurons is much different that the role of SP signaling. This should be considered throughout the manuscript.

This was reformulated in the revised text. We also made sure to clearly state that we are modulating Tac1-expressing neurons rather than SP signaling.

Line 129: An injection volume of 300nl seems much too large to achieve specific targeting of the pre-BotC. The authors should comment on this and describe how specificity was achieved in their experiments. Also, what was infused at 80nl/min during insertion of the needle? What type of needle was used?

As we demonstrated in Supplementary Figures 1 and 4, ChETA-eYFP expression induced by viral transduction spread beyond the targeted region. Expression seems moderate in nuclei involved in breathing and is more prominent in the preBötC. Based on the 3D Monte Carlo simulation, laser with power of 5mW (which is sufficient to stimulate breathing) spreads rostrocaudally by 0.470 mm. However, the combination of local ChETA-eYFP expression and focused light in the preBotC ensures that Tac1 preBotC neurons play a significant role in breathing.

While the cannula was lowered in the brain to inject AAV virus in the preBötC, we maintained a flow rate of 80nl/min therefore avoiding clogging of the cannula. *In-situ* hybridization show very little spread of virus (ChETA-eYFP expression) along the ventral/dorsal axis where the cannula was lowered. Needle specifications are mentioned in the revised text.

Line 297: What is baseline respiratory rate in anesthesia? What is anesthesia used? It would be helpful to mention this in Results. Is there a difference in viral expression at 2 vs 4 wks that explains the different results at these two timepoints?

Baseline respiratory rate ranged between 20-30 breaths/min. This is mentioned in Methods section. Values for every animal can be visualized in the Supplementary Figure 3 and will be available in the excel files of data.

We did not quantify viral expression for 2 and 4 weeks as *in-situ* hybridization of ChETA-eYFP expression at 2 weeks was limited (2 brains). Absence of response to photostimulation from all animals in this group was sufficient to support moving toward 4 weeks post-injection of virus. We did not aim to associate strength of response with the number of *Tac1* cells transfected by AAV.

Line 304: Breathing was significantly affected when normalized to baseline, but not when raw breathing frequency was compared. This should be noted in the results.

This is addressed in the revised text.

Line 346: Figure 2e,f – The images seem to show clear examples of cells that express EYFP that DO NOT express Tac1… If so, this either suggests that the ISH for Tac1 is not labelling all Tac1 neurons, or the viral vector is not 100% specific for Tac1 neurons. Please comment.

As Tac1 mRNA is not confined to the cellular membrane like ChETA-eYFP, labelling of cells according to both mRNA can be deceptive. Through labelling and quantification of cells from *in-situ* hybridization, we determined that cells needed to express 4 or more dots and/or 1 cluster of mRNAs overlapping with or adjacent to the cell nucleus to be considered as a *Tac1* neuron. We included in Supplementary Figure 3 a close up of neurons showing that all cells marked for eYFP (green) also have more than 4 dots of *Tac1* mRNAs (red).

Line 430: suggest alternative to "in vivo". Studies in anesthetized mice also considered in vivo.

This has been modified in the revised text.

Line 431: RE the distinction between calm and active "states": "Based on baseline respiratory rate" – please elaborate on this here. How can baseline breathing rate be more or less than 125% baseline breathing rate? This seems somewhat arbitrary and needs justification. Since the only criteria is respiratory rate, if the data are separated in this way, the terminology used should reflect this. It would be more appropriate to remove "calm or active" throughout the manuscript and more appropriately state that the respiratory effects were frequency dependent, based upon the respiratory frequency at the time of stimulation, etc. Moreover, this analysis seems misleading since it is not surprising that the stimulations appear to have a larger effect on breathing when baseline frequency was slow (calm state) when the effect on breathing is quantified as a % change from baseline. To further illustrate this – since mice have a maximum frequency they can breathe, there is likely a ceiling effect. E.g. if the mouse is already breathing close to that maximum at baseline (active state), it would not be surprising that stimulation would be unable to increase it further.

This is a valid point. To better justify the distinction between "active" and "calm" states, we additionally used the video tracking of the freely-behaving animal during photostimulation. Animal was considered in "active" state when velocity (cm traveled per second) was over 0.1 cm/s and when activity (% of pixel changed in frame) was over 0.4%. During most of recoding when animal is immobile and calm, values for velocity and activity are under these two thresholds. This is addressed in the revised text.

Line 437: The raw frequency values are not included in the supplemental figures as stated in the text. This should be included.

This was removed from the revised text as raw frequency values are already shown in Figure 5.

Line 442: Again, this may not be surprising if mice have a maximum frequency they can breathe. E.g. if they are already breathing close to that maximum at baseline, stimulation will be unable to increase it further.

We mention in the revised text that this can be caused by a maximum rate of breathing reached by the animal limiting the effect of photostimulation.

Why were stimulations delivered in 200 ms "bouts"?

If the reviewer refers to the bouts of 20ms pulses, we based this stimulation on previous work including Alsahaﬁ et al., 2015 and Ausborn et al., 2018.

Figure 5- what about other breath parameters – e.g. amplitude/volume that seem to change with stimulations in awake mice in addition to frequency?

Amplitude/volume of breath in freely behaving animal did not significantly change following photosimulation. While we did not add the plot in Figure 5, this result is mentioned in the revised text with the respective statistics.

Figure 6d,e, – velocity and activity should not be expressed as a % baseline since in both cases the baseline value is ~0. Raw values would be more appropriate i.e. cm/s.

Velocity and activity in Figure 6d,e are shown in raw value in the revised figures.

Line 483-486: cannot make this conclusion based on the data presented, because control mice did not move fast or breath fast during the recordings. Therefore, the data do not demonstrate whether or not fast breathing and movement are also correlated in control mice.

This is a valid point. Relationship between the two behaviors in control animal has been removed from the revised text.

Line 496: Should 38% be considered a "small fraction"?

This has been modified in the revised text.

Discussion Line 588: "Tac1 role on breathing depends on the state of the animal (calm vs. active), and therefore the overall excitability of the respiratory network, as a minimal effect of Tac1 stimulation was observed when the animal was behaviorally active." The claim of state-dependence does not seem justified. The distinction between calm and active states has little rationale and is based only on respiratory frequency.

With additional criteria to define the behavioral states (body movement and activity), the definitions of calm and active do not depend on breathing as initially mentioned. This is addressed in the revised discussion.

Line 696: Entrainment of breathing by Tac1 neurons is not demonstrated here.

This is a valid point. As entrainment of breathing is not clearly demonstrated in freely behaving animals, this has been modified in the revised discussion.

[Editors’ note: what follows is the authors’ response to the second round of review.]Reviewer #3 (Recommendations for the authors):The authors have made important revisions to the manuscript and have included additional data that has improved the study. But, the paper would still benefit from additional revisions.Title – In this reviewer's opinion the data presented do not describe a "circuit" but characterize the function of a group of neurons in a single location. This may be better reflected by a title such as "Medullary tachykinin precursor 1 neurons contribute to rhythmic breathing".

We changed the title to: Medullary tachykinin precursor 1 neurons promote rhythmic breathing

Line 49-50: It seems somewhat misleading to state that a "new population" has been identified. As the authors state elsewhere, these neurons are very likely the same glutamatergic neurons that have been previously manipulated and functionally described using Vglut2-cre, Dbx1-cre, etc. The new insight is that some of these neurons express Tac1 and manipulating them has very similar consequences for breathing as manipulating the larger glutamatergic preBotC population.

We modified the wording accordingly.

Line 73: This is still overstated. There is very little data that substance P is "pivotal" for respiratory rhythm, and this could be confusing for the reader as it conflates the role of the neurons with the role of the neuropeptide. NK1-expressing neurons are pivotal, but this does not indicate a critical role for substance P.

We removed “pivotal” and changed wording accordingly.

Line 151-152: "Tac1 cells were only stimulated after virus incubated for 4 weeks, but not 2 weeks (n=25)" Do you mean it was only effective at 4 weeks?

We clarified this sentence. The effects of photostimulation were only observed in animals incubated with AAV for 4 weeks. No effect was observed in animals incubated for 2 weeks.

Line 442-458: This section is somewhat confusing and again conflates the function of substance p/NK1-r signaling with the function of glutamatergic preBotC neurons that express Tac1.

We modified the sentence accordingly and do not emphasize the role of substance P.

In general, the writing could be further improved. There are numerous typos, inconsistent use of tense, sentences with missing words, repetitions, etc. For example, the impact statement includes "neurons in the medulla" twice within the same sentence.

We thoroughly read the manuscript and correctly all tenses and repetitions in the manuscript.

It appears the reference list needs updating since some citations in the text do not appear in the bibliography.

We checked carefully all references. They are now all in the bibliography.